# Reliable Algorithm Selection for Machine Learning-Guided Design

**Clara Fannjiang** [1]   **Ji Won Park** [1]

## Abstract

Algorithms for machine learning-guided design, or *design algorithms*, use machine learning-based predictions to propose novel objects with desired property values. Given a new design task—for example, to design novel proteins with high binding affinity to a therapeutic target—one must choose a design algorithm and specify any hyperparameters and predictive and/or generative models involved. How can these decisions be made such that the resulting designs are successful? This paper proposes a method for *design algorithm selection*, which aims to select design algorithms that will produce a distribution of design labels satisfying a user-specified success criterion—for example, that at least ten percent of designs' labels exceed a threshold. It does so by combining designs' predicted property values with held-out labeled data to reliably forecast characteristics of the label distributions produced by different design algorithms, building upon techniques from prediction-powered inference (Angelopoulos et al., 2023). The method is guaranteed with high probability to return design algorithms that yield successful label distributions (or the null set if none exist), if the density ratios between the design and labeled data distributions are known. We demonstrate the method's effectiveness in simulated protein and RNA design tasks, in settings with either known or estimated density ratios.

## 1 Design Algorithm Selection

Machine learning-guided design aims to propose novel objects, or *designs*, that exhibit desired values of a property of interest by consulting machine learning-based predictions of the property in place of costly and time-consuming measurements. The approach has been used to design novel enzymes

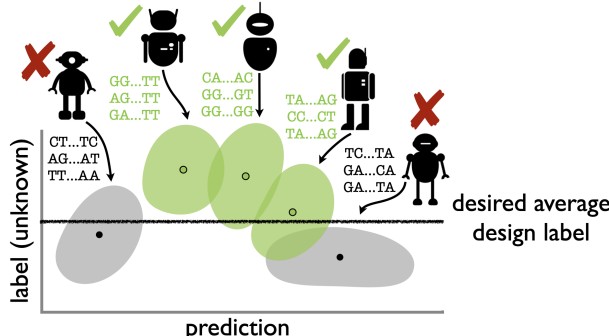

*Figure 1.* **Design algorithm selection.** Different design algorithm configurations (shown as robots) produce different distributions of designs (e.g., biological sequences). Distributions of designs' predicted property values (x-axis) and labels (y-axis) also differ (blobs). Before the costly step of acquiring design labels, design algorithm selection aims to choose design algorithm configurations that will satisfy a success criterion—for example, that the average design label surpasses a threshold (horizontal black line). This is challenging because designs' predictions can be misleading; true labels can be low even if predictions are high (e.g., rightmost blob).

that efficiently catalyze reactions of interest (Greenhalgh et al., 2021), photoreceptors with unprecedented light sensitivity for optogenetics (Bedbrook et al., 2019), and antibodies with enhanced binding affinity to therapeutic targets (Gruver et al., 2023), among many other applications. The methods used in such efforts, which we call *design algorithms*, are varied. Some entail sampling from a generative model that upweights objects with promising predictions (Brookes et al., 2019; Biswas et al., 2021); others start with initial candidates and iteratively introduce modifications that yield more desirable predictions (Bryant et al., 2021; Thomas et al., 2025). To propose designs, one must choose among these design algorithms. Many have consequential hyperparameter(s) that must be set, such as those that navigate a trade-off between departing from training data to achieve unprecedented predicted values and staying close enough that those predictions can be trusted (Biswas et al., 2021; Gruver et al., 2023; Zhu et al., 2024). One must also specify the predictive model that the design algorithm consults, as well as any generative model it may involve. These decisions collectively yield a *design algorithm configuration*, or *configuration* for short—for example, the design algorithm `AdaLead` (Sinai et al., 2020) using the hyperpa-

---

[1]Prescient Design, Genentech. Correspondence to: Clara Fannjiang <wong-fannjiang.clara@gene.com>.

*Proceedings of the 42nd International Conference on Machine Learning*, Vancouver, Canada. PMLR 267, 2025. Copyright 2025 by the author(s).

rameter settings $r = 0.2, \mu = 0.02, \kappa = 0.05$ and a ridge regression predictive model. The choice of the configuration dictates the resulting designs, and, consequently, whether the design effort succeeds or fails. Specifically, we say that a configuration is *successful* if it produces a distribution of design labels that satisfies a user-specified *success criterion*—for example, that at least ten percent of the design labels exceed some threshold. In this paper, we propose a solution to the problem of *design algorithm selection*:

> *How can one select a design algorithm configuration that is guaranteed to be successful?*

To anticipate whether a configuration will be successful, one can examine predicted property values for designs that it produces. However, these predictions can be particularly error-prone, as design algorithms often produce distributions of designs shifted away from the training data in order to achieve unprecedented predicted values (Fannjiang & Listgarten, 2024). We propose a method for design algorithm selection that combines predictions for designs with labeled data held out from training, in a way that corrects for these errors and enables theoretical guarantees on the selected configurations. First, we formalize design algorithm selection as a multiple hypothesis testing problem. We consider a finite set of candidate design algorithm configurations, called the *menu*, which we describe shortly. Each configuration on the menu is affiliated with a hypothesis test of whether it satisfies the success criterion. The method then computes a statistically valid p-value for this hypothesis test, by combining held-out labeled data with predicted property values for designs generated by the configuration. Loosely speaking, it uses the labeled data to characterize how prediction error biases a p-value based on predictions alone and removes this bias, building upon techniques from prediction-powered inference (Angelopoulos et al., 2023). Finally, the p-values are assessed with a multiple testing correction to choose a set of design algorithm configurations. The method is guaranteed with high probability to select successful configurations (or return the null set if none are identified), if the ratios between the design and labeled data densities are known for all configurations on the menu. If these density ratios are unknown, we show empirically that the method still frequently selects successful configurations using estimated density ratios.

The contents of the menu will depend on the task at hand. If a bespoke design algorithm has been developed specifically for the task, we may be interested in setting a key hyperparameter, such as real-valued hyperparameters that dictate how close to stay to the training data or other trusted points (Schubert et al., 2018; Linder et al., 2020; Biswas et al., 2021; Gruver et al., 2023; Fram et al., 2024; Tagasovska et al., 2024). The menu would then be a finite set of candidate values for that hyperparameter, such as a grid of values

between plausible upper and lower bounds. In other cases, such as when a variety of design algorithms are appropriate for the task, we may want to consider multiple options for the design algorithm, its hyperparameter(s), and the predictive model it consults. The menu could then comprise all combinations of such options. Indeed, the menu can include or exclude any configuration we desire; we can include options for any degree of freedom whose effect on the designs we want to consider, while holding fixed those that can be reliably set by domain expertise.

Our contributions are as follows. We introduce the problem of design algorithm selection, which formalizes a consequential decision faced by practitioners of machine learning-guided design and connects it to the goal of producing a distribution of design labels that satisfies a criterion. We propose a method for design algorithm selection, which combines designs' predicted property values with held-out labeled data in a principled way to reliably assess whether candidate design algorithm configurations will be successful. We provide theoretical guarantees for the configurations selected by the method, if the ratios between the design and labeled data densities are known. Finally, we demonstrate the method's effectiveness in simulated protein and RNA design tasks, including settings where these density ratios are unknown and must be estimated.

## 2 Problem Formalization

This section formalizes the design algorithm selection problem. The next section proposes our solution.

The goal of a design task is to find novel objects in some domain, $\mathcal{X}$, whose labels in some space $\mathcal{Y}$ satisfy a desired criterion. For example, one may seek protein sequences of length $L$, $x \in \mathcal{X} = \mathcal{A}^L$ where $\mathcal{A}$ is the set of amino acids, whose real-valued catalytic activities for a reaction of interest, $y \in \mathcal{Y} = \mathbb{R}$, surpass some threshold. Note that it is often neither necessary nor feasible for every proposed design to satisfy the criterion—it is sufficiently useful that some of them do so (Wheelock et al., 2022). This observation guides our formalization of the *success criterion*, which we describe shortly. We first describe the other components of our framework.

*Design algorithms* are methods that output novel objects whose labels are believed to satisfy the desired criterion. We focus on those that consult a predictive model, though our framework is agnostic as to how they do so; they may also use other sources of information, such as unlabeled data or biomolecular structures. An example of a design algorithm for the aforementioned enzyme design task is to sample from a generative model fit to sequences that are evolutionarily related to a known enzyme, and return the samples with desirable predictions (Thomas et al., 2025).

A *design algorithm configuration* or *configuration* is a specification of all the hyperparameter settings and models needed to deploy a design algorithm. Given a design task, a practitioner constructs a *menu*, $\Lambda$: a set of candidate configurations to be considered. For each configuration on the menu, $\lambda \in \Lambda$, we get predicted property values for $N$ designs produced by the configuration: $\{f_\lambda(x_i^\lambda)\}_{i=1}^N$, where $f_\lambda$ is the predictive model used by configuration $\lambda$.

We assume access to a set of i.i.d. *labeled data*: $x_i \sim P_{\text{lab}}$, $y_i \sim P_{Y|X=x}$, $i = 1, \ldots, n$, where $P_{\text{lab}}$ is the *labeled distribution* and $P_{Y|X=x}$ is the conditional distribution of the label random variable, $Y$, given the point $x$. We assume that this conditional distribution is fixed for every point in $\mathcal{X}$, as is the case when the label is dictated by the laws of nature. This data must be independent from the training data for the predictive models used on the menu, but it need not be from the training data distribution. Whenever unclear from context, we will say *held-out* or *additional* labeled data to disambiguate this data from the training data.

## 2.1 The Success Criterion

Given the above components, the goal of design algorithm selection is to select a subset of configurations, $\hat{\Lambda} \subseteq \Lambda$, that satisfy the *success criterion*, which we now formalize.

The designs produced by any design algorithm configuration, $\lambda \in \Lambda$, are sampled from some *design distribution* over $\mathcal{X}$, denoted $P_{X;\lambda}$. This distribution may be specified explicitly (Brookes et al., 2019; Zhu et al., 2024), or only implicitly, such as when the algorithm iteratively introduces mutations to training sequences based on the resulting predicted property values (Sinai et al., 2020). The design distribution in turn induces the *design label distribution* over $\mathcal{Y}$, denoted $P_{Y;\lambda}$, which is the distribution of design labels and can be sampled from as follows: $x \sim P_{X;\lambda}, y \sim P_{Y|X=x}$. Note that the labeled data and design data are related by covariate shift (Shimodaira, 2000): the distributions over $\mathcal{X}$, $P_{\text{lab}}$ and $P_{X;\lambda}$, differ, but the conditional distribution of the label for any point, $P_{Y|X=x}$ for any $x$, is fixed.

As previously noted, the aspiration for most design endeavors in practice is not that every single design satisfies a criterion, but that enough of them do so (Wheelock et al., 2022). Accordingly, our framework defines success in terms of the design label distribution, $P_{Y;\lambda}$, rather than the label of any specific design. The practitioner can specify any *success criterion* that requires the expected value of some function of the design labels to surpass some threshold:

$$\theta_\lambda := \mathbb{E}_{Y \sim P_{Y;\lambda}}[g(Y)] \geq \tau \qquad (1)$$

for some $g : \mathcal{Y} \to \mathbb{R}$ and $\tau \in \mathbb{R}$.[1] We call $\theta_\lambda$ the *population-*

---

[1]See §B.1 for generalization to other success criteria; here we focus on this special case for its broad applicability.

*level metric*. Examples include the mean design label when $g$ is the identity, as well as the fraction of designs whose labels exceed some value $\gamma \in \mathcal{Y}$, when $g(y) = \mathbb{1}[y \geq \gamma]$. For example, a practitioner can request that at least ten percent of the designs' labels exceed that of a wild type, $y_{\text{WT}}$, using the success criterion $\mathbb{E}_{Y \sim P_{Y;\lambda}}[\mathbb{1}[Y \geq y_{\text{WT}}]] \geq 0.1$. We call a configuration *successful* if it yields a design label distribution that achieves the success criterion.

Our goal is to select a subset of configurations from the menu such that, with guaranteed probability, every selected configuration is successful (or to return the empty set, if no successful configuration exists on the menu). That is, for any user-specified *error rate* $\alpha \in [0, 1]$, we aim to construct a subset $\hat{\Lambda} \subseteq \Lambda$ such that

$$\mathbb{P}(\theta_\lambda \geq \tau, \forall \lambda \in \hat{\Lambda}) \geq 1 - \alpha. \qquad (2)$$

If the density ratios between the design and labeled distributions are known for all configurations on the menu, our proposed method, which we detail next, guarantees Eq. 2.

## 3 Design Algorithm Selection by Prediction-Powered Multiple Testing

Our method tackles design algorithm selection by approaching it as a multiple hypothesis testing problem (Alg. 1). The goal is to select a subset of configurations that are all successful, such that the *error rate*—the probability of incorrectly including one or more unsuccessful configurations—is at most $\alpha$. To accomplish this, for each configuration on the menu, consider the null hypothesis that it is unsuccessful: $H_\lambda : \theta_\lambda := \mathbb{E}_{Y \sim P_{Y;\lambda}}[g(Y)] < \tau$. We compute a p-value, $p_\lambda$, for testing against this null hypothesis, as we describe shortly. Finally, we use the Bonferroni correction to select all configurations whose p-values are sufficiently small. This subset of configurations satisfies Eq. 2, our goal for design algorithm selection.

The key subproblem is obtaining statistically valid p-values for testing against the null hypotheses $H_\lambda, \lambda \in \Lambda$. These hypotheses concern the design label distributions, yet we do not have labels for the designs, only predictions. To extract information from these predictions without being misled by prediction error, we turn to prediction-powered inference (Angelopoulos et al., 2023), a framework that combines predictions with held-out labeled data to conduct valid statistical inference. Specifically, we use prediction-powered inference techniques adapted for covariate shift, due to the covariate-shift relationship between the design data and labeled data. We defer a thorough treatment to §A.1, but conceptually, the labeled data, weighted by the density ratios between the design and labeled distributions, is used to characterize how prediction error distorts estimation of the population-level metric, $\theta_\lambda$, based on predictions alone. This error characterization is then combined with the

predictions for the designs to compute p-values that have either asymptotic (Alg. 2) or finite-sample (Alg. 3) validity. Using the latter, the output of Algorithm 1 satisfies Eq. 2.

---

**Algorithm 1** Design algorithm selection by multiple hypothesis testing

---

**Inputs:** $N$ designs generated by each design algorithm configuration on the menu, $\{x_i^\lambda\}_{i=1}^N, \forall \lambda \in \Lambda$; predictive models used by each configuration, $\{f_\lambda\}_{\lambda \in \Lambda}$; labeled data, $\{(x_j, y_j)\}_{j=1}^n$; error rate, $\alpha \in [0, 1]$.
**Output:** Selected configurations, $\hat{\Lambda} \subseteq \Lambda$.

1: **for** $\lambda \in \Lambda$ **do**
2:    Predictions for designs, $\hat{y}_i^\lambda \leftarrow f_\lambda(x_i^\lambda), i \in [N]$.
3:    Predictions for labeled data, $\hat{y}_j \leftarrow f_\lambda(x_i), j \in [n]$.
4:    $p_\lambda \leftarrow \text{GETPVALUE}(\{\hat{y}_i^\lambda\}_{i=1}^N, \{(x_j, y_j, \hat{y}_j)\}_{j=1}^n)$
5: **end for**
6: $\hat{\Lambda} \leftarrow \{\lambda \in \Lambda : p_\lambda \leq \alpha/|\Lambda|\}$

---

**Algorithm 2** Prediction-powered p-value for testing $H_\lambda : \theta_\lambda := \mathbb{E}_{Y \sim P_{Y;\lambda}}[g(Y)] < \tau$

---

**Inputs:** Predictions for designs, $\{\hat{y}_i^\lambda\}_{i=1}^N$; labeled data and their predictions, $\{(x_j, y_j, \hat{y}_j)\}_{j=1}^n$.
**Output:** p-value, $P$.

1: $\hat{\mu} \leftarrow \frac{1}{N} \sum_{i=1}^N g(\hat{y}_i^\lambda)$
2: $w_j \leftarrow \text{DENSITYRATIO}(x_j), \; j = 1, \ldots, n$
3: $\hat{\Delta} \leftarrow \frac{1}{n} \sum_{j=1}^n w_j(g(y_j) - g(\hat{y}_j))$
4: $\hat{\theta} \leftarrow \hat{\mu} + \hat{\Delta}$
5: $\hat{\sigma}_{\text{pred}}^2 \leftarrow \frac{1}{N} \sum_{i=1}^N (g(\hat{y}_i^\lambda) - \hat{\mu})^2$
6: $\hat{\sigma}_{\text{err}}^2 \leftarrow \frac{1}{n} \sum_{j=1}^n \left( w_j[g(y_j) - g(\hat{y}_j)] - \hat{\Delta} \right)^2$
7: $P \leftarrow 1 - \Phi\left( \left( \hat{\theta} - \tau \right) / \sqrt{\frac{\hat{\sigma}_{\text{pred}}^2}{N} + \frac{\hat{\sigma}_{\text{err}}^2}{n}} \right)$

---

**Theorem 3.1.** *For any error rate $\alpha \in [0, 1]$, function $g : \mathcal{Y} \to \mathbb{R}$, and threshold $\tau \in \mathbb{R}$, Algorithm 1 using Algorithm 3 as the p-value subroutine returns a subset of configurations from the menu, $\hat{\Lambda} \subseteq \Lambda$, that satisfies*

$$\mathbb{P}(\theta_\lambda \geq \tau, \, \forall \lambda \in \hat{\Lambda}) \geq 1 - \alpha$$

*where $\theta_\lambda := \mathbb{E}_{Y \sim P_{Y;\lambda}}[g(Y)]$, and the probability is over random draws of the labeled data and the designs for all configurations on the menu.*

Using asymptotically valid p-values (Alg. 2) yields an asymptotic version of the guarantee (Thm. A.1). In experiments with known density ratios between the design and labeled distributions, we always achieved error rates under $\alpha$ even with asymptotically valid p-values. Since these are faster to compute, as they leverage closed-form representations of the asymptotic null distributions, we recommend using these in practice.

**Selecting zero or multiple configurations** Note that an error rate of zero can be trivially achieved by returning the empty set—that is, by not selecting any configuration. This is a legitimate outcome if there are no successful configurations on the menu, but otherwise, we will show empirically that our method also exhibits a high *selection rate*, or rate of returning nonempty sets. On the other hand, if our method selects multiple configurations then one can safely use any (or any mixture) of them with the same high-probability guarantee of success. In particular, one can further narrow down $\hat{\Lambda}$ using additional criteria—for example, picking the selected configuration that produces the most diverse designs, in order to hedge against unknown future desiderata.

**Density ratios between design and labeled distributions** Computing the p-values requires the density ratio between the design and labeled distributions, $p_{X;\lambda}(x_i)/p_{\text{lab}}(x_i)$, for every configuration and every labeled instance. Important settings in biological sequence design where the design density can be evaluated are when designs are sampled from autoregressive generative models (Shin et al., 2021) or when the design distribution is a product of independent categorical distributions per sequence site (Weinstein et al., 2022; Zhu et al., 2024). Labeled sequence data is often generated by adding random substitutions to wild-type sequences (Biswas et al., 2021; Bryant et al., 2021) or by recombining segments of several "parent" sequences (Romero et al., 2013; Bedbrook et al., 2019), in which case $p_{\text{lab}}$ is also explicitly known. Valid p-values can also be computed if both $p_{X;\lambda}$ and $p_{\text{lab}}$ are only known up to normalizing factors, such as when sequences are generated by Potts models (Russ et al., 2020; Fram et al., 2024) (§B.2). In other settings, however, these density ratios need to be approximated. In experiments with unknown density ratios, we use multinomial logistic regression-based density ratio estimation (Srivastava et al., 2023) and show that our method still empirically outperforms others in selecting successful configurations. Although Theorems 3.1 or A.1 no longer apply in this setting, they inform us what guarantees we can recover, to the extent that the density ratios are approximated well.

**When have we gone "too far?"** For each configuration, the further apart the design and labeled distributions are, the higher the variance of the density ratios that are used to weight the labeled data. This reduces the effective sample size of the labeled data in characterizing prediction error, which leads to higher uncertainty about the value of the population-level metric, $\theta_\lambda$. Consequently, even if the configuration is successful, it may not be selected if the design and labeled distributions are too far apart. Indeed, this is desirable behavior: the method essentially identifies where over $\mathcal{X}$ we lack sufficient statistical evidence of achieving the success criterion. It returns the empty set if the predictions and labeled data do not collectively provide adequate evidence that any configuration on the menu is successful.

Another factor to how many configurations are selected is the multiple testing correction, as it determines what counts as adequate evidence of success. A conceptual strength of framing design algorithm selection as a multiple testing problem is that any multiple testing procedure that controls family-wise error rate (FWER) can be used in Line 6 of Algorithm 1. Our instantiation uses the Bonferroni correction, as it does not require any assumptions about how different configurations are related, and it yielded reasonably high selection rates with menus of a few hundred configurations in experiments. However, one can also substitute FWER-controlling procedures that account for hierarchical (Bretz et al., 2009) or correlation structure (Dudoit et al., 2003) in the menu, which could yield less conservative multiplicity corrections and therefore higher selection rates.

## 4    Related Work

Design algorithm selection belongs to a body of work on managing predictive uncertainty in machine learning-guided design. Design algorithms can be constrained to stay close to the training data or trusted reference points, such that predictions remain trustworthy (Brookes et al., 2019; Linder et al., 2020; Biswas et al., 2021; Gruver et al., 2023; Tagasovska et al., 2024), and out-of-distribution detection methods can flag individual designs whose predictions are unreliable (Damani et al., 2023). A variety of methods have been used to quantify predictive uncertainty for individual designs in biomolecular design tasks (Greenman et al., 2025), including those based on ensembling (Scalia et al., 2020; Gruver et al., 2021), Gaussian processes (Hie et al., 2020; Tran et al., 2020), and evidential learning (Soleimany et al., 2021). In particular, conformal prediction techniques produce prediction sets for designs that also have frequentist-style guarantees: the sets contain the design labels with guaranteed probability, where the probability is over drawing designs from the design distribution (Fannjiang et al., 2022; Prinster et al., 2023). However, it is unclear how to invoke such statements to make decisions about which designs to use, as there is no guarantee regarding the prediction sets for any *specific* designs of interest—for example, those whose prediction sets satisfy a desired condition.

Moreover, quantifying uncertainty for individual designs is perhaps unnecessary for the goal of many design campaigns in practice. Success often does not necessitate that every design performs as desired, but only that sufficiently many do so, regardless of which specific ones (Wheelock et al., 2022). For example, Wang et al. (2022) develop a method that selects designs from a pool of individual candidates, such that the selected subset contains a desired expected number whose label surpasses a threshold. Conformal selection methods (Jin & Candès, 2022; 2023; Bai & Jin, 2024) combine ideas from conformal prediction and multiple testing to also select designs from a candidate pool, with guaranteed upper bounds on the false discovery rate, or expected proportion of selected designs whose label falls below a desired threshold. The setting of these methods differs from ours in that they assume access to a pool of candidates that are exchangeable—for example, candidates drawn i.i.d. from some distribution. However, depending on the application, it may not always be straightforward to narrow down a large design space $\mathcal{X}$ to a suitable pool of candidates to begin with, without selecting a design algorithm configuration with which to generate designs. The goal of our work is therefore to choose from a set of candidate design algorithm configurations, each of which will produce a different distribution of designs when deployed. Our approach aims to select configurations that will achieve a success criterion, which we formalize as a user-specified population-level metric, $\theta_\lambda$, surpassing some threshold.

Bayesian optimization (BO) (Shahriari et al., 2016) is a well-studied paradigm for iteratively selecting designs, acquiring their labels, and updating a predictive model in order to optimize a property of interest. In each round, using the language of our framework, BO prescribes a design algorithm: it chooses the design (or batch of designs (Desautels et al., 2014)) that globally maximizes some *acquisition function* that quantifies desirability based on the model's predictions, and which typically incorporates the model's predictive uncertainty, such as the expected improvement of a design's label over the best one found thus far. A typical goal of BO algorithms is to converge to the global optimum as the rounds progress, under regularity conditions on the property of interest (Srinivas et al., 2010; Bull, 2011; Berkenkamp et al., 2019). In contrast, the goal of design algorithm selection is to achieve criteria on the distribution of design labels to be imminently proposed. Such guarantees can help justify designs to stakeholders when acquiring labels for even one round is resource-intensive, and the priority is to find designs that achieve specific improvements within one or a few rounds, rather than to eventually find the best possible design as the rounds progress indefinitely. However, nothing precludes batch BO algorithms from being used as design algorithms in our framework: configurations of these algorithms with different hyperparameter settings or acquisition functions, for example, can be on the menu.

Similar to our work, Wheelock et al. (2022) also aim to characterize the design label distribution. To do so, however, their approach focuses on how to construct a *forecast*, or model of the conditional distribution of the label, for individual designs. An equally weighted mixture of designs' forecasts then serves as a model of the design label distribution. A rich body of work also exists on *calibrating* forecasts, such that various aspects of these conditional distributions are statistically consistent with held-out labeled data (Gneiting et al., 2007; Kuleshov et al., 2018; Song et al.,

2019). Our approach differs in that it directly estimates the population-level metric, $\theta_\lambda$, that determines whether a configuration is successful, rather than modeling the labels of individual designs. That is, our method handles prediction error in a way that is specifically tailored for the endpoint of selecting which design algorithm configuration to deploy.

At a technical level, our work uses prediction-powered inference techniques (Angelopoulos et al., 2023) adapted for covariate shift, in order to conduct statistically valid hypothesis tests of whether configurations are successful. The multiple hypothesis testing approach is similar to that of Angelopoulos et al. (2021), who use multiple testing to set hyperparameters for predictive models to achieve desired risk values with high-probability guarantees.

# 5  Experiments

We first demonstrate that our method selects successful configurations with high probability, as guaranteed by theory, when the design and labeled densities are known. Next, we show that it still selects successful configurations more effectively than alternative methods when these densities are estimated. Code for these experiments is at https://github.com/clarafy/design-algorithm-selection 🔗.

Two metrics are of interest: error rate and selection rate. Error rate is the empirical frequency at which a method selects a configuration that fails the success criterion (Eq. 1), over multiple trials of sampling designs from each configuration as well as held-out labeled data (for methods that require it). Selection rate is the empirical frequency over those same trials at which a method selects anything at all. A good method achieves a low error rate while maintaining a high selection rate, which may be challenging for ambitious success criteria.

Note that the prediction-only and `GMMForecasts` methods do not need held-out labeled data. For fair comparison, we ran these methods with predictive models trained on the total amount of labeled data used by our method (10k instances, whereas our method trained on 5k and held out 5k).

**Prediction-only method** This baseline uses only the predictions for a configuration's designs to assess whether it is successful. Specifically, it follows the same multiple testing framework as our method, but computes the p-values using the designs' predictions as if they were labels (Alg. 6).

**Gaussian-mixture-model forecasts method** Both this and the next method construct a forecast, or model of the conditional distribution of the label, for every design generated by a configuration. They then use the equally weighted mixture of the designs' forecasts as a model of the design label distribution for a configuration, and select the configuration if this model satisfies the success criterion. These two methods differ in how they construct the forecast for a designed sequence; we now describe the first, called `GMMForecasts`.

We follow Wheelock et al. (2022), who model the conditional distribution of the label as a mixture of two Gaussians with sequence-specific parameters, which capture beliefs over the label if the sequence is "functional" or "nonfunctional." After training this model, we infer forecasts for every design produced by a configuration $\lambda$, with different values of a hyperparameter $q \in [0, 1]$ that controls, roughly speaking, how much the forecasts deviate from the training data (see §C.2 for details). The equally weighted mixture of these forecasts, $P_\lambda^{\text{GMM}}$, serves as a model of the design label distribution for configuration $\lambda$. We select $\lambda$ if $\mathbb{E}_{Y \sim P_\lambda^{\text{GMM}}}[g(Y)] \geq \tau$.

**Calibrated forecasts method** For this method, called `CalibratedForecasts`, the forecast for each design is initially modeled as a Gaussian with mean and variance set to the predictive mean and variance, respectively, given by the predictive model. We then use the labeled data to *calibrate* these forecasts post hoc using isotonic regression (Kuleshov et al., 2018) (see §C.3 for details). We select configuration $\lambda$ if $\mathbb{E}_{Y \sim P_\lambda^{\text{cal}}}[g(Y)] \geq \tau$, where $P_\lambda^{\text{cal}}$ is the equally weighted mixture of the calibrated forecasts for the designs from $\lambda$.

**Conformal prediction method** We adapted conformal prediction techniques to conduct design algorithm selection (see §C.4 for details). This method has similar theoretical guarantees to our method, but is prohibitively conservative: it never selected anything in any of our experiments. We therefore exclude these results for clarity of visualization.

## 5.1  Algorithm Selection for Designing Protein GB1

The design task for the first set of experiments was to design novel protein sequences that have high binding affinity to an immunoglobulin, by specifying the amino acids at four particular sites of a protein called GB1. These experiments simulate library design, an important practical setting in which both the design and labeled densities have closed-form expressions. Specifically, the most time- and cost-effective protocols today for synthesizing protein sequences in the wet lab can be described mathematically as sampling the amino acid at each site independently from a site-specific categorical distribution, whose parameters we can set; the density for any sequence is then the product of the probabilities of the amino acid at each site. We follow the design algorithm developed by Zhu, Brookes, & Busia *et al.* (2024): after training a predictive model of binding affinity (see §D for details), $f$, we set the parameters of the site-specific categorical distributions such that sequences with high predictions have high likelihood, as follows.

Let $\mathcal{Q}$ denote the class of distributions that are products of four independent categorical distributions over twenty

amino acids. The authors use stochastic gradient descent to approximately solve $q_\lambda = \arg\min_{q \in \mathcal{Q}} D_{\mathrm{KL}}(p_\lambda^\star \,||\, q)$, where $p_\lambda^\star(x) \propto \exp(f(x)/\lambda)$ and $\lambda > 0$ is a temperature hyperparameter that needs to be set carefully. Note that the training distribution, described shortly, was similar to a uniform distribution, which corresponds to $\lambda = \infty$. If $\lambda$ is low, designs sampled from $q_\lambda$ tend to have predictions that are high but untrustworthy since $q_\lambda$ is far from the training distribution, while the opposite is true for high $\lambda$ (Fig. 2). The goal is therefore to select $\lambda$ such that $q_\lambda$ is successful.

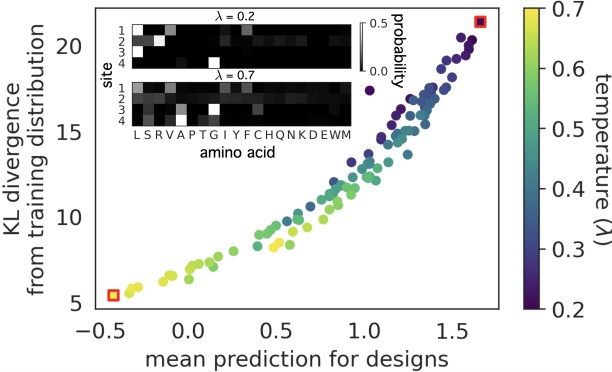

*Figure 2.* **Library design for protein GB1.** Mean prediction and KL divergence from the training distribution of the 101 design algorithm configurations on the menu. Each dot corresponds to the design distribution, $q_\lambda$, for a specific value of the temperature hyperparameter, $\lambda$. Two red-outlined squares correspond to design distributions for the lowest temperature ($\lambda = 0.2$) and the highest temperature ($\lambda = 0.7$), whose parameters values are shown in the inset (top and bottom heatmaps, respectively).

**Labeled data** To simulate training and held-out labeled data, we sampled sequences from a common baseline called the NNK library, which is close to uniform categorical distributions at every site, but slightly modified to reduce the probability of stop codons. For the labels, we used a data set that contains experimentally measured binding affinities for every sequence in $\mathcal{X}$ (Wu et al., 2016)—that is, all $20^4$ variants of protein GB1 at 4 specific sites. Labels were log-ratios relative to wild-type GB1, whose label was 0.

**Menu and success criteria** The menu contained 101 values of $\lambda$ between 0.2 and 0.7. We used the following success criteria: that the mean design label surpasses $\tau$, for $\tau \in [0, 1.5]$ (for reference, the mean training label was $-4.8$), and that the exceedance over 1 (i.e., the fraction of design labels that exceed 1, using $g(Y) = \mathbf{1}[Y \geq 1]$) surpasses $\tau$, for $\tau \in [0, 1]$ (for reference, the training labels' exceedance over 1 was 0.006).

**Selection experiments** For the prediction-only method and `GMMForecasts`, which do not need held-out labeled data, we trained the binding affinity predictive model on 10k

labeled sequences. We then solved for $q_\lambda$, as described above, for all $\lambda \in \Lambda$. For each of $T = 10$ trials, we sampled $N = 1\text{M}$ designs from each $q_\lambda$ and ran both methods to select temperatures.

For our method and `CalibratedForecasts`, which use held-out labeled data, we trained the predictive model on 5k labeled sequences and solved for $q_\lambda, \lambda \in \Lambda$. For each of $T = 500$ trials, we sampled $n = 5\text{k}$ additional labeled sequences, which were used to run both methods along with $N = 1\text{M}$ designs from each $q_\lambda$.

We can compute the true value of $\theta_\lambda$ for all $\lambda \in \Lambda$, since we have labels for all sequences in $\mathcal{X}$ (Wu et al., 2016). The error rate for each method was then computed as $\sum_{t=1}^{T} \mathbb{1}[\exists \lambda \in \hat{\Lambda} \text{ s.t. } \theta_\lambda < \tau]/T$. We used $\alpha = 0.1$ as a representative value for the desired error rate.

**Results** A good design algorithm selection method achieves a low error rate and a high selection rate for a variety of success criteria (settings of $g$ and $\tau$ in the criterion $\mathbb{E}_{Y \sim P_{Y;\lambda}}[g(Y)] \geq \tau$). For success criteria concerning the mean design label (i.e., $g$ is the identity), the prediction-only and `CalibratedForecasts` methods had error rates of 100% for most values of $\tau$ considered (Fig. 3a). Particularly for the former, the mean design label achieved by selected temperatures could be considerably lower than $\tau$ (Fig. 3b).

Our method had error rates below the desired level of $\alpha$ for all values of $\tau$ considered. The selection rate was 100% for a broad range of $\tau$, though it gradually declined for $\tau > 1$, reflecting increasing conservativeness for more stringent success criteria (Fig. 3a). We also ran our method with estimated density ratios for comparison, with very similar results (see §D.1 and Fig. 6).

For `GMMForecasts`, smaller values of the hyperparameter $q \in [0, 1]$ yield forecasts that are, roughly speaking, more similar to the training data (see §C.2 for details). Using $q \in \{0, 0.5\}$ was prohibitively conservative and never selected anything on any trial. Using the maximum value, $q = 1$, did yield high selection and low error rates for $\tau < 0.5$. However, for greater values of $\tau$, the method ceased selecting anything, incorrectly indicating that no configuration can achieve these success criteria (Fig. 3a). All methods had similar qualitative performance for success criteria concerning the exceedance over 1, though `GMMForecasts` was slightly less conservative (Fig. 7).

Recall that both the prediction-only and `GMMForecasts` methods do not require held-out labeled data, and therefore used predictive models trained on all 10k labeled sequences. Our method outperformed them in spite of using only half the amount of training data, demonstrating the benefit in this setting of reserving labeled data for quantifying and managing the consequences of prediction error.

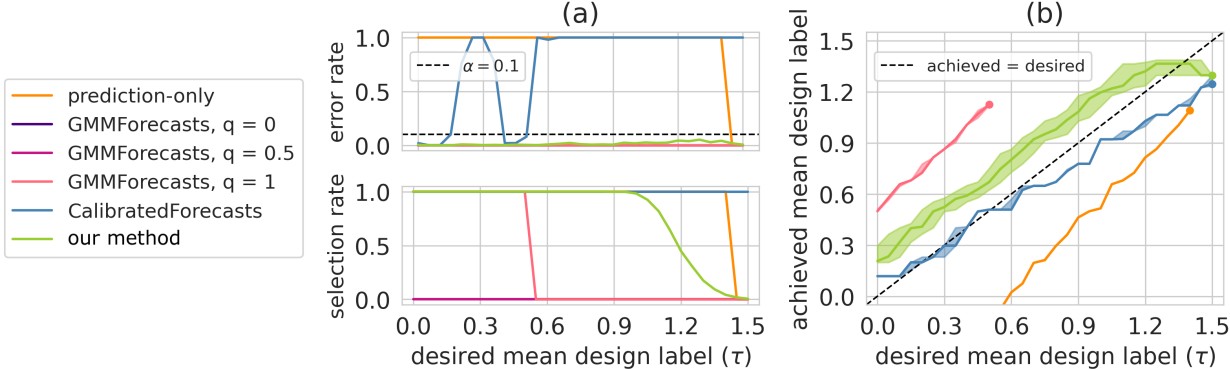

*Figure 3.* **Design algorithm selection for designing protein GB1.** (a) Error rate (top; lower is better) and selection rate (bottom; higher is better) for all methods, for range of values of $\tau$, the desired mean design label. For reference, the mean label of the training data was $-4.8$. `GMMForecasts` with hyperparameter $q \in \{0, 0.5\}$ (dark and medium purple lines) never selected anything, resulting in error and selection rates of zero for all $\tau$. (b) For each method, the median (solid line) and $20^{\text{th}}$ to $80^{\text{th}}$ percentiles (shaded regions) of the lowest mean design label achieved by selected configurations, over trials for which the method did not return the empty set. Dots mark where each median trajectory ends (i.e., the value of $\tau$ beyond which a method ceases to select any configuration, and the lowest mean design label of configurations selected for that $\tau$). Results on or above the dashed diagonal line indicate that selected configurations are successful.

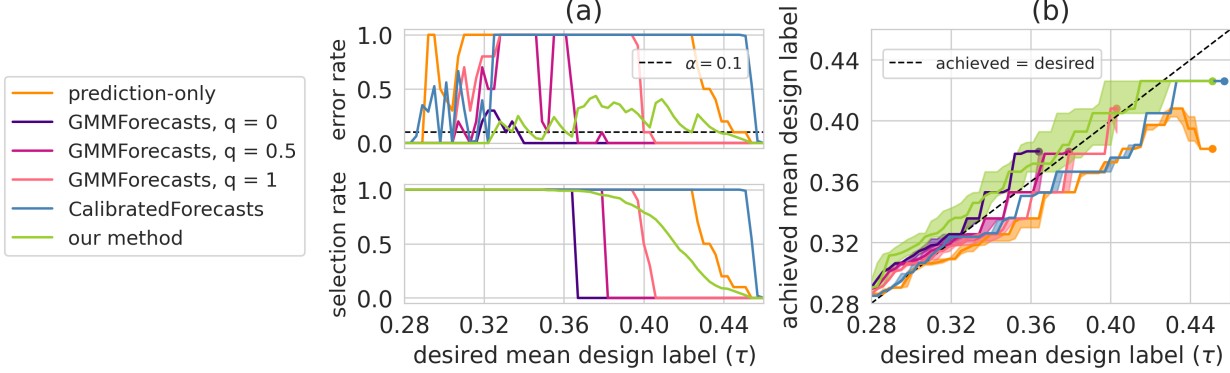

*Figure 4.* **Design algorithm selection for designing RNA binders.** (a) Error rate (top; lower is better) and selection rate (bottom; higher is better) for all methods, for range of values of $\tau$, the desired mean design label. For reference, the mean label of the training data was 0.28. (b) For each method, the median (solid line) and $20^{\text{th}}$ to $80^{\text{th}}$ percentiles (shaded regions) of the lowest mean design label achieved by selected configurations, over trials for which the method did not return the empty set. Dots mark where each median trajectory ends (i.e., the value of $\tau$ beyond which a method ceases to select any configuration, and the lowest mean design label of configurations selected for that $\tau$). Results on or above the dashed diagonal line indicate that selected configurations are successful.

### 5.2 Algorithm Selection for Designing RNA Binders

These next experiments show the utility of our method in a setting that involves a variety of design algorithms and predictive models on the menu, and that requires density ratio estimation. The task is to design length-50 RNA sequences that bind well to an RNA target, where the label is the ViennaFold binding energy (Lorenz et al., 2011).

**Labeled data** To simulate training and held-out labeled data, we generated random mutants of a "seed" sequence, with 0.08 probability of mutation at each site. Each mutant, $x$, was assigned a noisy label, $y = \text{BINDINGENERGY}(x) + \epsilon$, where $\epsilon \sim \mathcal{N}(0, \sigma = 0.02)$.

**Menu and success criteria** The menu contained the following design algorithms and respective hyperparameter settings (see §E.1 and Figs. 8, 9 for details): AdaLead (Sinai et al., 2020), with its threshold hyperparameter, $\kappa$, set to values in $[0.2, 0.01]$; the algorithm used by Biswas et al. (2021), which approximately runs MCMC sampling from $p(x) \propto \exp(f(x)/T)$, with the temperature hyperparameter, $T$, set to values in $[0.005, 0.02]$; Conditioning by Adaptive Sampling (Brookes et al., 2019) and Design by Adaptive Sampling (Brookes & Listgarten, 2018), with their quantile hyperparameter, $Q$, set to values in $[0.1, 0.9]$; and Proximal Exploration (Ren et al., 2022) with default hyperparameters, as the authors did not highlight any critical hyperparameters. Each of these design algorithms and respective hyperparam-

eter settings was run with three different predictive models, resulting in a menu of 78 configurations: a ridge regression model, with the regularization hyperparameter set by leave-one-out cross-validation, an ensemble of fully connected neural networks, and an ensemble of convolutional neural networks. The success criterion was that the mean design label surpasses $\tau$, for $\tau \in [0.28, 0.5]$ (for comparison, the mean training label was 0.28). We also ran our method with an expanded menu of 249 configurations involving an additional hyperparameter and architectures, to assess how the multiple testing correction affects selection rates for larger menus (see §E.1 and Fig. 8).

**Density ratio estimation** Since the configurations do not have closed-form design densities, we used multinomial logistic regression-based density ratio estimation (MDRE) (Srivastava et al., 2023), which trains a classifier between designs and labeled sequences to estimate the density ratios between their distributions (see §E.2 for details).

**Selection experiments** For the prediction-only method and `GMMForecasts`, which do not require held-out labeled data, we first trained the three predictive models on 10k labeled sequences. For each of $T = 10$ trials, we sampled $N = 50k$ designs from each configuration and ran both methods to select configurations.

For our method and `CalibratedForecasts`, which use held-out labeled data, we trained the three predictive models on 5k labeled sequences. These sequences, as well as $N = 50k$ designs generated from each configuration, were used to fit the MDRE model used by our method. For each of $T = 200$ trials, we sampled $n = 5k$ additional labeled sequences, which were used to run both methods along with the $N$ designs from each configuration.

For each configuration, $\lambda$, we took the average of 500k design labels to serve as $\theta_\lambda$. The error rate for each method was then computed as $\sum_{t=1}^{T} \mathbb{1}[\exists \lambda \in \hat{\Lambda} \text{ s.t. } \theta_\lambda < \tau]/T$, and we used $\alpha = 0.1$ as the desired error rate.

**Results** The prediction-only and `CalibratedForecasts` methods had 100% error rates for much of the range of $\tau$ considered. Our method had much lower error rates, though greater than $\alpha$ for $\tau > 0.32$ due to density ratio estimation error (Fig. 4a, top). Furthermore, when our method selected configurations that were unsuccessful, their mean design labels were still close to $\tau$ (Fig. 4b; the shaded green region does not extend far below the dashed diagonal line). Our method assesses configurations more accurately at the cost of selected subsets, $\hat{\Lambda}$, that are higher variance (Fig. 4b; the shaded green region is wider than other shaded regions), due to the reduced effective sample size of the weighted labeled data.

`GMMForecasts` with $q = 0$ had low error rates, even zero, for $\tau < 0.37$, but was the most conservative method: it

stopped selecting any configurations for greater $\tau$ (Fig. 4a, bottom). Using $q \in \{0.5, 1\}$ maintained 100% selection rates for broader ranges of $\tau$ at the cost of much higher error rates, though like our method, the errors' consequences were not severe, especially for $q = 0.5$: the mean design labels of unsuccessful configurations were not far below $\tau$ (Fig. 4b).

Overall, compared to any alternative method except `GMMForecasts` with $q = 0$, our method had lower error rates over the range of $\tau$ for which the alternative had non-zero selection rates (Fig. 4a). Our method also maintained non-zero selection rates for a broader range of $\tau$ than `GMMForecasts` with any $q$ (Fig. 4a, bottom). These results—which hold even with estimated density ratios, and holding out labeled data—illustrate the benefits of quantifying the consequences of prediction error on downstream quantities or decisions of interest, over focusing on the uncertainties of individual predictions.

## 6 Discussion

We introduced an algorithm selection method for machine learning-guided design, which selects design algorithm configurations that will be successful with high probability—that is, produce a distribution of design labels that satisfies a user-specified population-level success criterion. It does so by using held-out labeled data to characterize and then undo how prediction error biases the assessment of whether a configuration is successful. Though the present work focuses on success in a single "round" of design, it can also provide a principled decision-making framework for multi-round design endeavors in which a top priority is that the designs at each round achieve certain criteria—for example, to justify resources for acquiring their labels.

As with other uncertainty quantification methods that achieve frequentist-style guarantees under covariate shift (Tibshirani et al., 2019; Fannjiang et al., 2022; Prinster et al., 2023; Jin & Candès, 2023), the method uses the density ratios between the design and labeled distributions. Advances in density ratio estimation techniques will strengthen the method's performance in settings where these density ratios are not known—in particular, techniques that are tailored for importance-weighted mean estimation. Another promising direction is the incorporation of multiple testing procedures that respect structure among configurations on the menu, which may enable less conservative selection than the Bonferroni correction. Looking forward, we encourage continued work on how to address predictive uncertainty in machine learning-guided design with respect to how it directly impacts endpoints or decisions of interest, rather than with general-purpose notions of uncertainty (Greenman et al., 2025).

## Acknowledgements

Our gratitude goes to Anastasios N. Angelopoulos, Stephen Bates, Richard Bonneau, Kyunghyun Cho, Andreas Loukas, Ewa Nowara, Stephen Ra, Samuel Stanton, Nataša Tagasovska, and Tijana Zrnic for helpful discussions and feedback on this work.

## Impact Statement

This work enables practitioners of machine learning-guided design to select successful algorithms more reliably than existing methods. There are many potential societal consequences of machine learning-guided design, none which we feel must be specifically highlighted here.

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

# A    Additional Algorithms and Proofs

---

**Algorithm 3** Prediction-powered p-value for testing $H_\lambda : \theta_\lambda := \mathbb{E}_{Y \sim P_{Y;\lambda}}[g(Y)] < \tau$ (finite sample-valid)

---

**Inputs:** Predictions for designs, $\{\hat{y}_i^\lambda\}_{i=1}^N$; labeled data and their predictions, $\{(x_j, y_j, \hat{y}_j)\}_{j=1}^n$; small grid spacing, $\delta > 0$.
**Output:** p-value, $P$.

1:  $w_j \leftarrow \text{DENSITYRATIO}(x_j), \ j = 1, \dots, n$
2:  **for** $\alpha \in \{0, \delta, \dots, 1 - \delta, 1\}$ **do**
3:      $L_\alpha \leftarrow \text{PPMEANLB}(\alpha, \ \{\hat{y}_i^\lambda\}_{i=1}^N, \ \{(y_j, \hat{y}_j, w_j)\}_{j=1}^n)$
4:      **if** $L_\alpha > \tau$ **then**
5:          $P \leftarrow \alpha$
6:          **break**
7:      **end if**
8:  **end for**

---

**Algorithm 4** PPMEANLB: Prediction-powered confidence lower bound on $\theta_\lambda := \mathbb{E}_{Y \sim P_{Y;\lambda}}[g(Y)]$ (finite sample-valid)

---

**Inputs:** Significance level, $\alpha \in [0, 1]$; predictions for designs, $\{\hat{y}_i^\lambda\}_{i=1}^N$; labels, predictions, and density ratios for labeled data, $\{(y_i, \hat{y}_i, w_i)\}_{i=1}^n$; range, $[L, U]$, of $g(Y)$; bound, $D$, on the density ratios.
**Output:** Confidence lower bound, $L$.

1:  $\hat{\mu}_{\text{lower}} \leftarrow \text{MEANLB}(0.1 \cdot \alpha, \ \{g(\hat{y}_i^\lambda)\}_{i=1}^N, \ [L, U])$
2:  $\hat{\Delta}_{\text{lower}} \leftarrow \text{MEANLB}(0.9 \cdot \alpha, \ \{w_j \cdot (g(y_j) - g(\hat{y}_j))\}_{j=1}^n, \ [D(L - U), D(U - L)])$
3:  $L \leftarrow \hat{\mu}_{\text{lower}} + \hat{\Delta}_{\text{lower}}$

---

**Algorithm 5** MEANLB: Confidence lower bound on a mean (finite sample-valid; Waudby-Smith & Ramdas (2023))

---

**Inputs:** Significance level, $\alpha \in [0, 1]$; data, $\{z_i\}_{i=1}^n$; range of random variable, $[L, U]$.
**Output:** Confidence lower bound, $B$.

1:  $z_i \leftarrow (z_i - L)/(U - L), i = 1, \dots, n$
2:  $\mathcal{A} \leftarrow \{0, \delta, \dots, 1 - \delta, 1\}$
3:  $M_0^+(m) \leftarrow 1, m \in \mathcal{A}$
4:  **for** $t = 1, \dots, n$ **do**
5:      $\hat{\mu}_t \leftarrow \frac{0.5 + \sum_{i=1}^t z_i}{t+1}, \hat{\sigma}_t^2 \leftarrow \frac{0.25 + \sum_{i=1}^t (z_i - \hat{\mu}_t)^2}{t+1}, \lambda_t \leftarrow \sqrt{\frac{2 \log(2/\alpha)}{n \hat{\sigma}_{t-1}^2}}$
6:      **for** $m \in \mathcal{A}$ **do**
7:          $M_t^+(m) \leftarrow \left(1 + \min\{\lambda_t, \frac{0.5}{m}\}(z_t - m)\right) M_{t-1}^+(m)$
8:          **if** $M_t^+(m) \geq 1/\alpha$ **then**
9:              $\mathcal{A} \leftarrow \mathcal{A} \setminus \{m\}$
10:         **end if**
11:     **end for**
12: **end for**
13: $B \leftarrow \min \mathcal{A} \cdot (U - L) + L$

---

**Theorem A.1.** *For any error rate $\alpha \in [0, 1]$, function $g : \mathcal{Y} \to \mathbb{R}$, and threshold $\tau \in \mathbb{R}$, Algorithm 1 using Algorithm 2 as the p-value subroutine returns a subset of configurations from the menu, $\hat{\Lambda} \subseteq \Lambda$, that satisfies*

$$\liminf_{n,N \to \infty} \mathbb{P}(\theta_\lambda \geq \tau, \ \forall \lambda \in \hat{\Lambda}) \geq 1 - \alpha, \tag{3}$$

*where $\theta_\lambda := \mathbb{E}_{Y \sim P_{Y;\lambda}}[g(Y)]$, $n$ and $N$ are the amounts of labeled data and designs from each configuration, respectively, $\frac{n}{N} \to r$ for some $r \in (0, 1)$, and the probability is over random draws of the labeled data and the designs from all configurations on the menu.*

## A.1 Proofs of Theorems 3.1 and A.1

The proofs of Theorems 3.1 and A.1 rely on establishing the validity of the p-values computed by Algorithms 3 and 2, respectively. Both algorithms use the framework of prediction-powered inference (Angelopoulos et al., 2023), which constructs confidence intervals and p-values for an estimand of interest—here, the population-level metric, $\mathbb{E}_{Y \sim P_{Y;\lambda}}[g(Y)]$—by combining predictions for $Y$ with an estimand-specific characterization of prediction error called the *rectifier*. For our estimand, these two components emerge from the following simple decomposition of the population-level metric:

$$\mathbb{E}_{Y \sim P_{Y;\lambda}}[g(Y)] = \mathbb{E}_{X \sim P_{X;\lambda}}[g(f(X))] + \mathbb{E}_{X,Y \sim P_{X;\lambda} \cdot P_{Y|X}}[g(Y) - g(f(X))], \tag{4}$$

where $f$ is any predictive model. The first summand is the mean prediction for designs, and the second is the mean prediction bias, which serves as the rectifier. More generally, prediction-powered inference encompasses any estimand that can be expressed as the minimizer of an expected convex loss (see Appendix B.1), in which case the rectifier can be derived from a similar decomposition of the expected loss gradient.

Let $\{x_i^\lambda\}_{i=1}^N$ denote $N$ designs from configuration $\lambda$, and let $\{(x_j, y_j)\}_{j=1}^n$ denote the $n$ labeled instances. We can use the predictions for the designs to estimate the first summand in Eq. 4 as

$$\hat{\mu} := \frac{1}{N} \sum_{i=1}^N g(f(x_i^\lambda)), \tag{5}$$

and use the $n$ labeled instances to estimate the second, the rectifier. Specifically, we leverage the covariate shift relationship between the design and labeled data to rewrite the rectifier as

$$\mathbb{E}_{X,Y \sim P_{X;\lambda} \cdot P_{Y|X}}[g(Y) - g(f(X))] = \mathbb{E}_{X,Y \sim P_{\text{lab}} \cdot P_{Y|X}}\left[\frac{p_{X;\lambda}(X) p_{Y|X}(Y \mid X)}{p_{\text{lab}}(X) p_{Y|X}(Y \mid X)}(g(Y) - g(f(X)))\right]$$

$$= \mathbb{E}_{X,Y \sim P_{\text{lab}} \cdot P_{Y|X}}\left[\frac{p_{X;\lambda}(X)}{p_{\text{lab}}(X)}(g(Y) - g(f(X)))\right].$$

This final expression can be estimated using the labeled data, where each instance is weighted by the density ratio between the design and label distributions:

$$\hat{\Delta} := \frac{1}{n} \sum_{j=1}^n \frac{p_{X;\lambda}(x_j)}{p_{\text{lab}}(x_j)}(g(y_j) - g(f(x_j))). \tag{6}$$

Adding together the two estimates in Eqs. 5 and 6 yields a prediction-powered estimate of the population-level metric,

$$\hat{\theta} := \hat{\mu} + \hat{\Delta}, \tag{7}$$

from which a prediction-powered p-value can be computed. Specifically, Algorithm 2 follows standard protocol for constructing asymptotically valid p-values, by first deriving the asymptotic null distribution of the prediction-powered estimate, and then evaluating its survival function to produce a p-value. Alternatively, Algorithm 3 inverts finite sample-valid, prediction-powered confidence lower bounds on the population-level metric to obtain finite sample-valid p-values.

We now briefly describe why the prediction-powered estimate of the population-level metric, Eq. 7, generally has lower variance—and yields correspondingly more powerful p-values—than ignoring the $N$ predictions and using only the $n$ weighted labeled instances to estimate the population-level metric. Specifically, the covariate shift relationship between the design and labeled data allows us to rewrite the population-level metric as

$$\mathbb{E}_{Y \sim P_{Y;\lambda}}[g(Y)] = \mathbb{E}_{X,Y \sim P_{X;\lambda} \cdot P_{Y|X}}[g(Y)] = \mathbb{E}_{X,Y \sim P_{\text{lab}} \cdot P_{Y|X}}\left[\frac{p_{X;\lambda}(X) p_{Y|X}(Y \mid X)}{p_{\text{lab}}(X) p_{Y|X}(Y \mid X)}g(Y)\right]$$

$$= \mathbb{E}_{X,Y \sim P_{\text{lab}} \cdot P_{Y|X}}\left[\frac{p_{X;\lambda}(X)}{p_{\text{lab}}(X)}g(Y)\right], \tag{8}$$

and correspondingly construct a "labeled-only" estimate,

$$\frac{1}{n} \sum_{j=1}^n \frac{p_{X;\lambda}(x_j)}{p_{\text{lab}}(x_j)}g(y_j). \tag{9}$$

We now compare the asymptotic variances of the prediction-powered estimate, Eq. 7, and the labeled-only estimate, Eq. 9. Critically, we assume that $N \gg n$, as it is typically far cheaper to generate designs in silico than it is to acquire labeled data. The central limit theorem gives the asymptotic variance of the prediction-powered estimate as

$$\frac{1}{N}\mathrm{Var}(g(f(X^\lambda))) + \frac{1}{n}\mathrm{Var}\left(\frac{p_{X;\lambda}(X)}{p_{\mathrm{lab}}(X)}\left(g(Y) - g(f(X))\right)\right).$$

For $N \gg n$, this expression is dominated by the second term, the variance of the weighted prediction error scaled by $1/n$ (indeed, we can make the first term arbitrarily small by computationally generating more designs). The asymptotic variance of the labeled-only estimate is the variance of the weighted labels scaled by $1/n$,

$$\frac{1}{n}\mathrm{Var}\left(\frac{p_{X;\lambda}(X)}{p_{\mathrm{lab}}(X)}g(Y)\right).$$

So long as the variance of the weighted prediction error is smaller than that of the weighted labels—that is, the predictions explain some of the variance of labels—the asymptotic variance of the prediction-powered estimate will be smaller than that of the labeled-only estimate. That is, the prediction-powered estimate uses information from the predictions to increase its effective sample size compared to the labeled-only estimate.

**Proof of Theorem A.1 (asymptotically valid design algorithm selection)**

*Proof.* We first establish the asymptotic validity of the p-value, $p_\lambda$, computed by Algorithm 2. Specifically, we show that under the null hypothesis, $H_\lambda : \theta_\lambda < \tau$, $p_\lambda$ is the survival function of the asymptotic distribution of the prediction-powered estimate $\hat{\theta}$, and therefore satisfies

$$\limsup_{n,N\to\infty} \mathbb{P}(p_\lambda \leq u) \leq u, \ \forall u \in [0, 1], \tag{10}$$

where $n/N \to r$ for some $r \in (0, 1)$.

We first derive the asymptotic distribution of $\hat{\theta} := \hat{\mu} + \hat{\Delta}$, where $\hat{\mu} := (1/N)\sum_{i=1}^{N} g(\hat{y}_i^\lambda)$ is the average prediction for $N$ designs, and $\hat{\Delta} := (1/n)\sum_{j=1}^{n} w_j(g(y_j) - g(\hat{y}_j))$ where $w_j = p_{X;\lambda}(x_j)/p_{\mathrm{lab}}(x_j)$ is the importance-weighted average prediction bias for the $n$ labeled instances. The central limit theorem implies that

$$\sqrt{N}(\hat{\mu} - \mathbb{E}[\hat{\mu}]) \xrightarrow{d} \mathcal{N}(0, \sigma_{\mathrm{pred}}^2),$$

$$\sqrt{n}(\hat{\Delta} - \mathbb{E}[\hat{\Delta}]) \xrightarrow{d} \mathcal{N}(0, \sigma_{\mathrm{err}}^2),$$

where $\sigma_{\mathrm{pred}}^2 := \mathrm{Var}_{X\sim P_{X;\lambda}}[g(f(X))]$ and $\sigma_{\mathrm{err}}^2 := \mathrm{Var}_{X,Y\sim P_{X;\lambda}\cdot P_{Y|X}}[g(Y) - g(f(X))]$. Applying the continuous mapping theorem to the sum of $\hat{\mu}$ and $\hat{\Delta}$ yields that

$$\sqrt{N}(\hat{\mu} + \hat{\Delta} - \mathbb{E}[\hat{\mu} + \hat{\Delta}]) = \sqrt{N}(\hat{\mu} - \mathbb{E}[\hat{\mu}]) + \sqrt{\frac{N}{n}}\sqrt{n}(\hat{\Delta} - \mathbb{E}[\hat{\Delta}])$$

$$\xrightarrow{d} \mathcal{N}(0, \sigma_{\mathrm{pred}}^2 + \frac{1}{r}\sigma_{\mathrm{err}}^2)$$

where recall that $n/N \to r$. Since

$$\mathbb{E}[\hat{\mu} + \hat{\Delta}] = \mathbb{E}_{X\sim P_{X;\lambda}}[g(f(X))] + \mathbb{E}_{X,Y\sim P_{X;\lambda}\cdot P_{Y|X}}[g(Y) - g(f(X))] = \mathbb{E}_{Y\sim P_{Y;\lambda}}[g(Y)] := \theta_\lambda,$$

we equivalently have

$$\sqrt{N}(\hat{\mu} + \hat{\Delta} - \theta_\lambda) \xrightarrow{d} \mathcal{N}(0, \sigma_{\mathrm{pred}}^2 + \frac{1}{r}\sigma_{\mathrm{err}}^2).$$

We can now evaluate the survival function of this distribution under the null hypothesis, $H_\lambda : \theta_\lambda < \tau$, to obtain a p-value. Specifically,

$$p_\lambda = 1 - \Phi\left(\frac{\hat{\theta} - \tau}{\sqrt{\hat{\sigma}^2/N}}\right),$$

where $\hat{\sigma}^2$ is any consistent estimate of $\sigma_{\text{pred}}^2 + \frac{1}{r}\sigma_{\text{err}}^2$, satisfies Eq. 10. We use the estimate $\hat{\sigma}^2 = \hat{\sigma}_{\text{pred}}^2 + \frac{N}{n}\hat{\sigma}_{\text{err}}^2$ where $\hat{\sigma}_{\text{pred}}^2 := \frac{1}{N}\sum_{i=1}^{N}(g(\hat{y}_i^\lambda) - \hat{\mu})^2$ and $\hat{\sigma}_{\text{err}}^2 := \frac{1}{n}\sum_{j=1}^{n}(w_j[g(y_j) - g(\hat{y}_j)] - \hat{\Delta})^2$, which is consistent as $\hat{\sigma}_{\text{pred}}^2$ and $\hat{\sigma}_{\text{err}}^2$ are consistent estimates of $\sigma_{\text{pred}}^2$ and $\sigma_{\text{err}}^2$, respectively.

Having established the validity of $p_\lambda, \lambda \in \Lambda$, we can control the family-wise error rate (FWER) with a Bonferroni correction:

$$
\begin{aligned}
\limsup_{n,N\to\infty}\text{FWER} := \limsup_{n,N\to\infty}\mathbb{P}\left(\bigcup_{\lambda\in\Lambda:\,\theta_\lambda<\tau}\lambda\in\hat{\Lambda}\right) &\leq \sum_{\lambda\in\Lambda:\,\theta_\lambda<\tau}\limsup_{n,N\to\infty}\mathbb{P}\left(\lambda\in\hat{\Lambda}\right) \\
&= \sum_{\lambda\in\Lambda:\,\theta_\lambda<\tau}\limsup_{n,N\to\infty}\mathbb{P}\left(p_\lambda \leq \frac{\alpha}{|\Lambda|}\right) \\
&\leq |\{\lambda\in\Lambda:\,\theta_\lambda<\tau\}|\cdot\frac{\alpha}{|\Lambda|} \\
&\leq |\Lambda|\cdot\frac{\alpha}{|\Lambda|} = \alpha,
\end{aligned}
$$

where the first line uses a union bound, the second follows from the definition of $\hat{\Lambda}$ in Algorithm 1, and the third is due to the validity of each $p_\lambda$. This gives us $\liminf_{n,N\to\infty}\mathbb{P}(\theta_\lambda \geq \tau, \, \forall\lambda\in\hat{\Lambda}) = 1 - \limsup_{n,N\to\infty}\text{FWER} \geq 1 - \alpha$.

$\square$

**Proof of Theorem 3.1 (finite sample-valid design algorithm selection)**

*Proof.* We first show that the p-value computed by Algorithm 3, $p_\lambda$, has finite-sample validity, meaning that under the null hypothesis, $H_\lambda : \theta_\lambda < \tau$, we have $\mathbb{P}(p_\lambda \leq u) \leq u, \forall u \in [0,1]$. First, the confidence lower bound, $L_\alpha$, computed by PPMEANLB (Alg. 4) is valid: it satisfies $\mathbb{P}(\theta_\lambda \geq L_\alpha) \geq 1 - \alpha$. This follows from the fact that MEANLB produces valid confidence lower bounds, $\hat{\mu}_{\text{lower}}$ and $\hat{\Delta}_{\text{lower}}$, for $\mathbb{E}_{X\sim P_{X;\lambda}}[g(f(X))]$ and $\mathbb{E}_{X,Y\sim P_{X;\lambda}\cdot P_{Y|X}}[g(Y) - g(f(X))]$, respectively (Theorem 3 from Waudby-Smith & Ramdas (2023)). Adding together these bounds therefore yields a valid confidence lower bound, $L_\alpha$, for $\theta_\lambda$. Algorithm 3 then constructs a p-value by inverting $L_\alpha$:

$$p_\lambda = \inf\{\alpha \in [0,1] : L_\alpha \geq \tau\}. \tag{11}$$

This p-value is valid because under the null hypothesis, $H_\lambda : \theta_\lambda < \tau$, for all $u \in [0,1]$ we have

$$\mathbb{P}(p_\lambda \leq u) \leq \mathbb{P}(\theta_\lambda < L_u) = 1 - \mathbb{P}(\theta_\lambda \geq L_u) \leq 1 - (1-u) = u,$$

where the first inequality follows from the definition of $p_\lambda$ in Eq. 11 and the fact that $\theta_\lambda < \tau$, and the second inequality comes from the validity of $L_u$.

Having established the validity of $p_\lambda, \lambda \in \Lambda$, the family-wise error rate (FWER), or the probability that one or more unsuccessful configurations is selected, can be controlled with the Bonferroni correction:

$$
\begin{aligned}
\text{FWER} := \mathbb{P}\left(\bigcup_{\lambda\in\Lambda:\,\theta_\lambda<\tau}\lambda\in\hat{\Lambda}\right) &\leq \sum_{\lambda\in\Lambda:\,\theta_\lambda<\tau}\mathbb{P}\left(\lambda\in\hat{\Lambda}\right) \\
&= \sum_{\lambda\in\Lambda:\,\theta_\lambda<\tau}\mathbb{P}\left(p_\lambda \leq \frac{\alpha}{|\Lambda|}\right) \\
&\leq |\{\lambda\in\Lambda:\,\theta_\lambda<\tau\}|\cdot\frac{\alpha}{|\Lambda|} \\
&\leq |\Lambda|\cdot\frac{\alpha}{|\Lambda|} = \alpha,
\end{aligned}
$$

where the first line uses a union bound, the second follows from the definition of $\hat{\Lambda}$ in Algorithm 1, and the third is due to the validity of each $p_\lambda$. We then have $\mathbb{P}(\theta_\lambda \geq \tau, \, \forall\lambda\in\hat{\Lambda}) = 1 - \text{FWER} \geq 1 - \alpha$.

$\square$

# B   Extensions

## B.1   More General Success Criteria

The main text considers success criteria of the following form: $\theta_\lambda := \mathbb{E}_{Y \sim P_{Y;\lambda}}[g(Y)] \geq \tau$ for some $g : \mathcal{Y} \to \mathbb{R}, \tau \in \mathbb{R}$. More generally, we can use prediction-powered inference techniques to compute valid p-values whenever the population-level metric, $\theta_\lambda$, can be expressed as the minimizer of the expectation of some convex loss (Angelopoulos et al., 2023):

$$\theta_\lambda := \arg\min_\theta \mathbb{E}_{X,Y \sim P_{X;\lambda} \cdot P_{Y|X}}[\ell_\theta(X,Y)],$$

where $\ell_\theta$ is convex in $\theta$. When $\ell_\theta(X,Y) = (g(Y) - \theta)^2$ for some $g : \mathcal{Y} \to \mathbb{R}$, we recover the special case in the main text. We could not conceive of practical settings requiring this general characterization of $\theta_\lambda$, but it may be useful for future work.

## B.2   Design and Labeled Densities Known up to Normalizing Constants

We can compute asymptotically valid p-values if we have unnormalized forms of the design and labeled densities, such as when sequences are generated from energy-based models (Biswas et al., 2021), Potts models (Russ et al., 2020; Fram et al., 2024), or other Markov random fields. Specifically, assume we can evaluate $p_{X;\lambda}^{\text{u}}(x) = a \cdot p_{X;\lambda}(x)$ and $p_{\text{lab}}^{\text{u}}(x) = b \cdot p_{\text{lab}}(x)$ for unknown constants $a, b \in \mathbb{R}$, where the superscript indicates that the densities are unnormalized. To leverage these in place of the exact densities in Algorithm 2, consider the corresponding scaled density ratios on the labeled data, $w_j^{\text{u}} := p_{X;\lambda}^{\text{u}}(x_j)/p_{\text{train}}^{\text{u}}(x_j), j = 1, \ldots, n$. The self-normalized importance-weighted estimator of prediction bias,

$$\hat{\Delta}^{\text{u}} := \frac{\sum_{j=1}^n w_j^{\text{u}} \cdot (g(y_j) - g(\hat{y}_j))}{\sum_{j=1}^n w_j^{\text{u}}},$$

is a consistent estimator of the rectifier in Eq. 4, $\mathbb{E}_{X,Y \sim P_{X;\lambda} \cdot P_{Y|X}}[g(Y) - g(f(X))]$. Since $\hat{\Delta}^{\text{u}}$ is a ratio of means, the delta method can be used to derive its asymptotic variance (Owen, 2013), which can be estimated as

$$\hat{\sigma}_{\hat{\Delta}^{\text{u}}}^2 := \frac{1}{n} \frac{\frac{1}{n} \sum_{j=1}^n (w_j^{\text{u}})^2 \cdot ([g(y_j) - g(\hat{y}_j)] - \hat{\Delta}^{\text{u}})^2}{\left(\frac{1}{n} \sum_{j=1}^n w_j^{\text{u}}\right)^2}.$$

We can then compute an asymptotically valid p-value using Algorithm 2, but replacing $\hat{\Delta}$ with $\hat{\Delta}^{\text{u}}$ in Line 3 and $\hat{\sigma}_{\text{err}}^2/n$ with $\hat{\sigma}_{\hat{\Delta}^{\text{u}}}^2$ in Line 7.

# C   Other Methods

## C.1   Prediction-Only Method

---

**Algorithm 6** Prediction-only p-value

---

**Inputs:** Predictions for designs, $\{\hat{y}_i^\lambda\}_{i=1}^N$; desired threshold, $\tau \in \mathbb{R}$.
**Output:** p-value, $P$.

1: $\hat{\theta} \leftarrow \frac{1}{N} \sum_{i=1}^N g(\hat{y}_i^\lambda)$
2: $\hat{\sigma}_{\text{pred}}^2 \leftarrow \frac{1}{N} \sum_{i=1}^N (g(\hat{y}_i^\lambda) - \hat{\theta})^2$
3: $P \leftarrow 1 - \Phi\left((\hat{\theta} - \tau)/\sqrt{\hat{\sigma}_{\text{pred}}^2/N}\right)$

---

The prediction-only method runs multiple testing (Alg. 1) with p-values computed using only the predictions for the designs (Alg. 6), treating them as if they were labels. These p-values are asymptotically valid for testing whether $\mathbb{E}_{X \sim P_{X;\lambda}}[g(f(X))] \geq \tau$—that is, whether the expected function of *predictions* for designs, but not necessarily their labels, surpasses a threshold.

## C.2   Gaussian-Mixture-Model Forecasts Method

The `GMMForecasts` method follows Wheelock et al. (2022), who model the forecast for a designed sequence as a mixture of two Gaussians (representing beliefs over the label if the sequence is "nonfunctional" and "functional") with sequence-specific

mixture proportion, means, and variances. To construct these forecasts, their approach first assumes access to a predictive mean and variance for each designed sequence, as described below. It then uses the training data to fit a mapping from these initial predictive means and variances to Gaussian mixture model (GMM) parameters. It also seeks to address covariate shift between the design and training data by using a sequence's edit distance from a reference training sequence (set to wild-type GB1 in the GB1 experiments, and the seed sequence in the RNA binder experiments) as an additional feature in fitting this mapping. Forecasts—that is, GMM parameters—are then inferred for each designed sequence.

The forecasts also involve a key hyperparameter, $q \in [0, 1]$. For each designed sequence, after GMM parameters are inferred, the mean of the "functional" Gaussian is adjusted by taking a convex combination of it and the original predictive mean, where $q$ weights the latter. Using different values of $q$ reflects how much one trusts the predictions; high values result in forecasts where the "functional" Gaussian mean is determined largely by the original predictive mean. We ran the method with $q \in \{0, 0.5, 1\}$ to span the range of possible values.

**Predictive mean and variance for a sequence** When using predictive models that were ensembles (i.e., the fully connected ensemble in the GB1 experiments, and the fully connected and convolutional ensembles in the RNA binder experiments), the predictive mean and variance for a sequence was set to the empirical mean and variance, respectively, of the predictions for that sequence. When using the ridge regression model in the RNA binder experiments, the model weights were fit on $90\%$ of the training data, and the predictive mean for a sequence was set to the model's prediction for it. The (homogeneous) predictive variance was set to the model's mean squared error over the remaining $10\%$ of the training data.

### C.3   Calibrated Forecasts Method

The `CalibratedForecasts` method is based on the idea that forecasts, or models of the conditional distributions of the label, should be statistically consistent with held-out labeled data. Specifically, we want the forecasts to be *calibrated* as defined by Kuleshov et al. (2018), whose definition is related to the notion of probabilistic calibration (Gneiting et al., 2007):

$$P_{X,Y}(Y \le F_X^{-1}(p))) = p, \ \forall p \in [0, 1], \tag{12}$$

where $F_X$ denotes the CDF of the forecast for a sequence $X \in \mathcal{X}$, and the probability is over the distribution of labeled data. That is, for any $p \in [0, 1]$, the label falls under the $p$-quantile given by a calibrated forecast with frequency $p$.

For a given sequence, let an initial forecast be a Gaussian with mean and variance set to a predictive mean and variance, as described above for `GMMForecasts`. We then use the held-out labeled data to learn a transformation of these initial forecast CDFs, such that the transformed CDF is closer to achieving calibration (Eq. 12). Specifically, we use isotonic regression, following Kuleshov et al. (2018). We then construct a forecast for every designed sequence by first forming the initial forecast, and then transforming the corresponding CDF with the fitted isotonic regression function.

### C.4   Conformal Prediction Method

---

**Algorithm 7** Conformal prediction-based method for design algorithm selection

---

**Inputs:** Designs generated with each configuration, $\{x_i^\lambda\}_{i=1}^N$ for all $\lambda \in \Lambda$; predictive models used by each configuration, $\{f_\lambda\}_{\lambda \in \Lambda}$; held-out labeled data, $\{(x_j, y_j)\}_{j=1}^n$; desired threshold, $\tau \in \mathbb{R}$; error rate, $\alpha \in [0, 1]$.
**Output:** Subset of selected configurations, $\hat{\Lambda} \in \Lambda$.

1: **for** $\lambda \in \Lambda$ **do**
2:    Predictions for designs, $\hat{y}_i^\lambda \leftarrow f_\lambda(x_i^\lambda), i = 1, \dots, N$
3:    Predictions for labeled data, $\hat{y}_j \leftarrow f_\lambda(x_i), j = 1, \dots, n$
4:    Density ratios for designs, $v_i \leftarrow \text{DENSITYRATIO}_\lambda(x_i^\lambda), i = 1, \dots, N$
5:    Density ratios for labeled data, $w_j \leftarrow \text{DENSITYRATIO}_\lambda(x_j), j = 1, \dots, n$
6:    $l_i^\lambda \leftarrow \text{SPLITCONFORMALLB}((\hat{y}_i^\lambda, v_i), \{(y_j, \hat{y}_j, w_j)\}_{j=1}^n, \alpha/(|\Lambda| \cdot N)), i = 1, \dots, N$
7:    $L_\lambda \leftarrow (1/N) \sum_{i=1}^N l_i^\lambda$
8: **end for**
9: $\hat{\Lambda} \leftarrow \{\lambda \in \Lambda : L_\lambda \ge \tau\}$

---

We adapt conformal prediction techniques to conduct design algorithm selection (Algs. 7, 8). For simplicity, we describe the method with $g$ as the identity function; for other functions, one can replace all references to labels with $g(Y)$.

---

**Algorithm 8** SPLITCONFORMALLB: split conformal lower bound for a design label

---

**Inputs:** a design's prediction and density ratio, $(\hat{y}^\lambda, v)$; labels, predictions, and density ratios for held-out labeled data, $\{(y_j, \hat{y}_j, w_j)\}_{j=1}^n$; $\alpha \in [0, 1]$.
**Output:** Lower bound, $L \in \mathbb{R}$.

1: $u_j \leftarrow \frac{w_j}{\sum_{j=1}^n w_j + v}, j = 1, \ldots, n$
2: $u \leftarrow \frac{v}{\sum_{j=1}^n w_j + v}$
3: $r \leftarrow (1 - \alpha)$-quantile of the distribution comprising the mixture of point masses $\sum_{j=1}^n u_j \cdot \delta_{\hat{y}_j - y_j} + u \cdot \delta_\infty$
4: $L \leftarrow \hat{y}^\lambda - r$

---

For a given configuration, $\lambda$, we can construct a valid lower bound for the empirical average of design labels, $(1/N) \sum_{i=1}^N y_i^\lambda$, by averaging Bonferroni-corrected conformal lower bounds for each design. Concretely, we can use a split conformal method (Tibshirani et al., 2019) to construct lower bounds, $l_i$, for the labels of $N$ designs, with confidence of $1 - \alpha/N$ each (Alg. 8). These lower bounds each satisfy $\mathbb{P}(y_i^\lambda \geq l_i) \geq 1 - \alpha/N$, where the probability is over random draws of the held-out labeled data. The average of these bounds, $L = (1/N) \sum_{i=1}^N l_i$, then satisfies $\mathbb{P}((1/N) \sum_{i=1}^N y_i^\lambda \geq L) \geq 1 - \alpha$. This is because the event $\{y_i^\lambda \geq l_i, \ \forall i \in [N]\}$ occurs with probability at least $1 - \alpha$ due to the Bonferroni correction, and on this event, we have $(1/N) \sum_{i=1}^N y_i^\lambda \geq (1/N) \sum_{i=1}^N l_i$.

To conduct design algorithm selection, we introduce an additional Bonferroni correction for the size of the menu, $|\Lambda|$: for each configuration, we construct a split conformal lower bound with a confidence of $1 - \alpha/(|\Lambda| \cdot N|)$ for each design, then take their average lower bound, $L_\lambda$. The event $\{(1/N) \sum_{i=1}^N y_i^\lambda \geq L_\lambda, \ \forall \lambda \in \Lambda\}$ occurs with probability at least $1 - \alpha$, which in turn implies that for $\hat{\Lambda} := \{\lambda \in \Lambda : L_\lambda \geq \tau\}$, we have

$$\mathbb{P}\left(\frac{1}{N} \sum_{i=1}^N y_i^\lambda \geq \tau, \ \forall \lambda \in \hat{\Lambda}\right) \geq 1 - \alpha.$$

Note that the lower bounds $L_\lambda, \lambda \in \Lambda$, are immensely conservative: because conformal prediction techniques are meant to characterize individual labels, they cannot naturally account for how prediction errors over many designs can "cancel out" in estimation of the mean design label. We rely instead on Bonferroni corrections to guarantee the extremely stringent criterion that every individual design's lower bound is correct, which in practice meant that $L_\lambda$ was always negative infinity in our experiments. Conformal prediction is fundamentally not the right tool when one is interested in how prediction error affects distributions of labels, rather than the labels of individual instances.

# D Protein GB1 Experiment Details

Labels were log enrichments relative to wild-type GB1, such that values greater (less) than 0 indicate greater (less) binding affinity than the wild type. Following Zhu, Brookes, & Busia *et al.* (2024), the predictive model, $f$, was an ensemble of fully connected neural networks, trained using a weighted maximum likelihood method that accounted for the estimated variance of each sequence's log enrichment label. After training on 5k labeled sequences, the model's predictions for all $x \in \mathcal{X}$ yielded an RMSE of 1.02, Pearson correlation coefficient of 0.79, and Spearman correlation coefficient of 0.68 (Fig. 5).

## D.1 Estimated Density Ratios

We ran our method with estimated density ratios, for success criteria concerning the mean design label. Specifically, we separately estimated the labeled distribution and the design distribution corresponding to each configuration. For the former, we performed maximum-likelihood estimation with Laplace smoothing (with pseudocounts of 1) to estimate the site-specific categorical distributions using the held-out labeled sequences; for the latter, we did the same using the designed sequences. For a given sequence, we then took the ratio of its densities under these two estimated distributions as the estimated density ratio. The results from our method using these estimated density ratios were very similar to the original results using the known density ratios (Fig. 6).

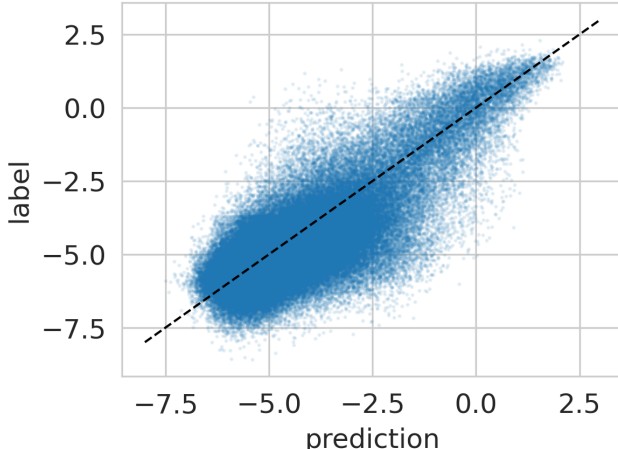

*Figure 5.* **Binding affinity predictions for all protein GB1 variants.** Predicted and measured binding affinity for all variants of protein GB1 at four sites of interest. Labels are from Wu et al. (2016) and predictions are from an ensemble of fully connected models trained on 5k labeled sequences from an NNK library. Both axes are log enrichments relative to wild-type GB1, whose label is 0.

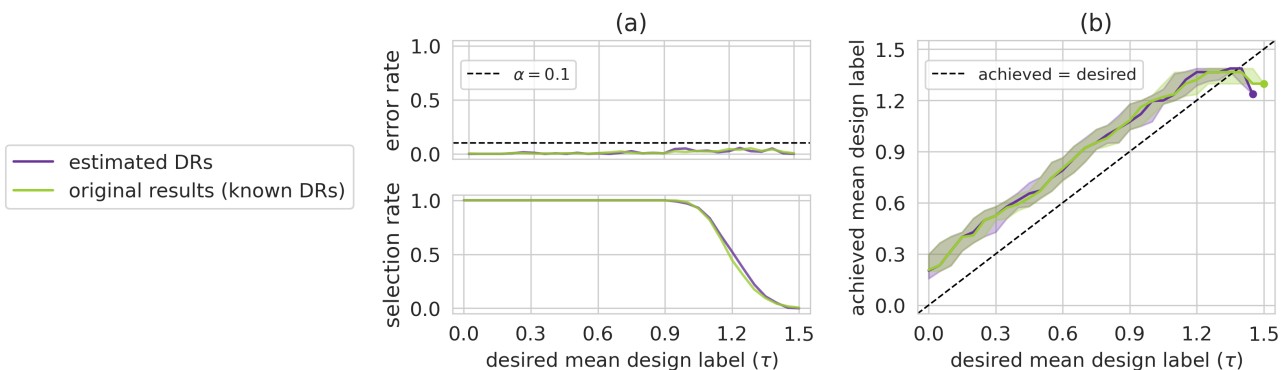

*Figure 6.* **Design algorithm selection for designing protein GB1 (estimated density ratios).** (a) Error rate (top; lower is better) and selection rate (bottom; higher is better) for our original results (with known density ratios between the designed and labeled distributions; green) and our method with estimated density ratios (purple). (b) The median (solid line) and $20^{th}$ to $80^{th}$ percentiles (shaded regions) of the lowest mean design label achieved by selected configurations, over trials for which the method did not return the empty set. Dots mark where each median trajectory ends (i.e., the value of $\tau$ beyond which the method ceased to select any configuration, and the lowest mean design label of configurations selected for that $\tau$). Results on or above the dashed diagonal line indicate that selected configurations are successful.

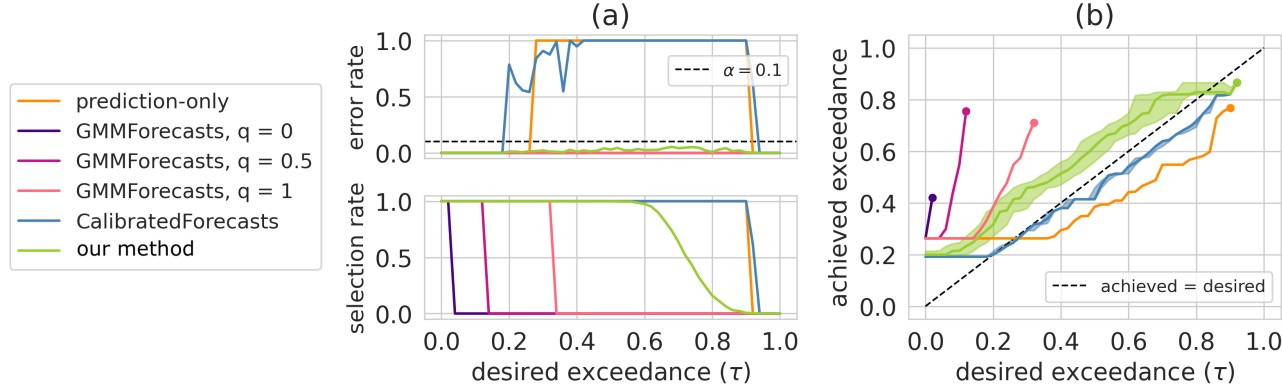

*Figure 7.* **Design algorithm selection for designing protein GB1 (exceedance-based success criteria).** (a) Error rate (top; lower is better) and selection rate (bottom; higher is better) for all methods, for range of values of $\tau$, the desired exceedance over 1 (that is, the fraction of designs whose label exceeds 1). For reference, the training labels' exceedance over 1 was 0.006. `GMMForecasts` with hyperparameter $q \in \{0, 0.5, 1\}$ (dark purple, medium purple, and pink lines) had error rates of zero for all values of $\tau$. (b) Median (solid line) and 20th to 80th percentiles (shaded regions), of the lowest mean design label achieved by selected configurations, across trials for which each method did not return the empty set. Dots mark where each median trajectory ends (i.e., the value of $\tau$ beyond which a method ceases to select any configuration, and the lowest mean design label of configurations selected for that $\tau$). Results on or above the dashed diagonal line indicate that selected configurations are successful.

## D.2 Exceedance-Based Success Criteria

We ran the same experiments described in §5.1 and shown in Figure 3, except with success criteria involving the exceedance over 1: $\mathbb{E}_{Y \sim P_{Y;\lambda}}[\mathbb{1}(Y \geq 1)] \geq \tau$, for $\tau \in [0, 1]$. For context, a label value of 1 was greater than 99.4% of the training labels and represents a binding affinity of about 2.7 times that of wild-type GB1. The results were qualitatively similar to those in the main text, except that `GMMForecasts` with $q \in \{0, 0.5\}$ was slightly less conservative and yielded high selection rates for a limited range of $\tau$ (Fig. 7).

## E RNA Binder Experiment Details

To facilitate interpretability of the label values, following Sinai et al. (2020) we normalized the ViennaFold binding energy (Lorenz et al., 2011) by that of the complement of the RNA target sequence, which can be seen as an estimate of the energy of the true optimal binding sequence. Consequently, a label value of 1 means a binding energy equal to that of the complement sequence.

### E.1 Menu of Design Algorithm Configurations

See Fig. 8 for a diagram of the menu structure.

**Predictive models** The three predictive models were a ridge regression model, where the ridge regularization hyperparameter was set by leave-one-out cross-validation; an ensemble of three fully connected neural networks, each with two 100-unit hidden layers; and an ensemble of three convolutional neural networks, each with three convolutional layers with 32 filters, followed by two 100-unit hidden layers. Each model in both ensembles was trained for five epochs using Adam with a learning rate of $10^{-3}$.

**Design algorithm hyperparameter settings** For `AdaLead` (Sinai et al., 2020), the values of the threshold hyperparameter on the menu were $\kappa \in \{0.2, 0.15, 0.1, 0.05, 0.01\}$. For `Biswas`, the algorithm used by Biswas et al. (2021), the values of the temperature hyperparameter on the menu were $T \in \{0.02, 0.015, 0.01, 0.005\}$. For Conditioning by Adaptive Sampling (Brookes et al., 2019), or `CbAS`, the values of the quantile hyperparameter on the menu were $Q \in \{0.1, 0.2, \ldots, 0.9\}$. For Design by Adaptive Sampling (Brookes & Listgarten, 2018) (`DbAS`) with either the fully connected or convolutional models, the values of the quantile hyperparameter on the menu were $Q \in \{0.1, 0.2, \ldots, 0.9\}$, and with ridge regression,

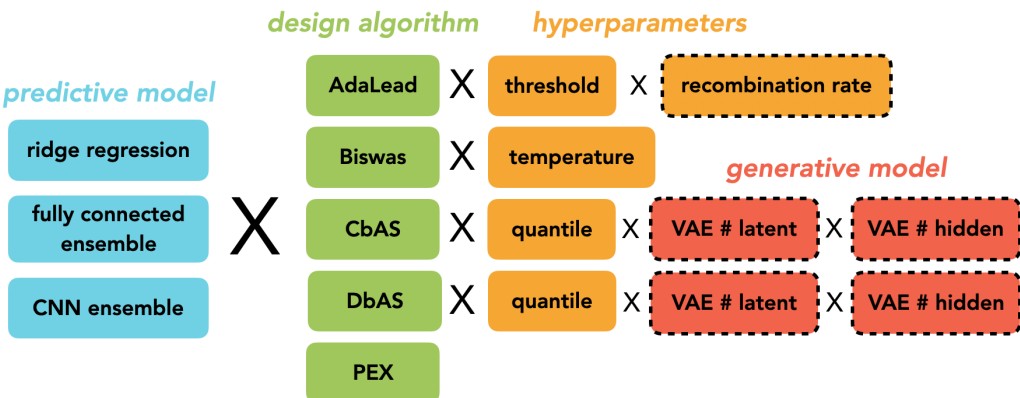

*Figure 8.* **Menu structure for designing RNA binders.** The menu for the results in the main text (size 78) had a nested configuration space involving categorical options for the predictive model and design algorithm, and grids of real values for one hyperparameter for each design algorithm. The expanded menu for the results in Figure 10 (size 249) adds an additional real-valued hyperparameter for `AdaLead`, and integer-valued options for the generative model architecture for `CbAS` and `DbAS` (dash-outlined cells).

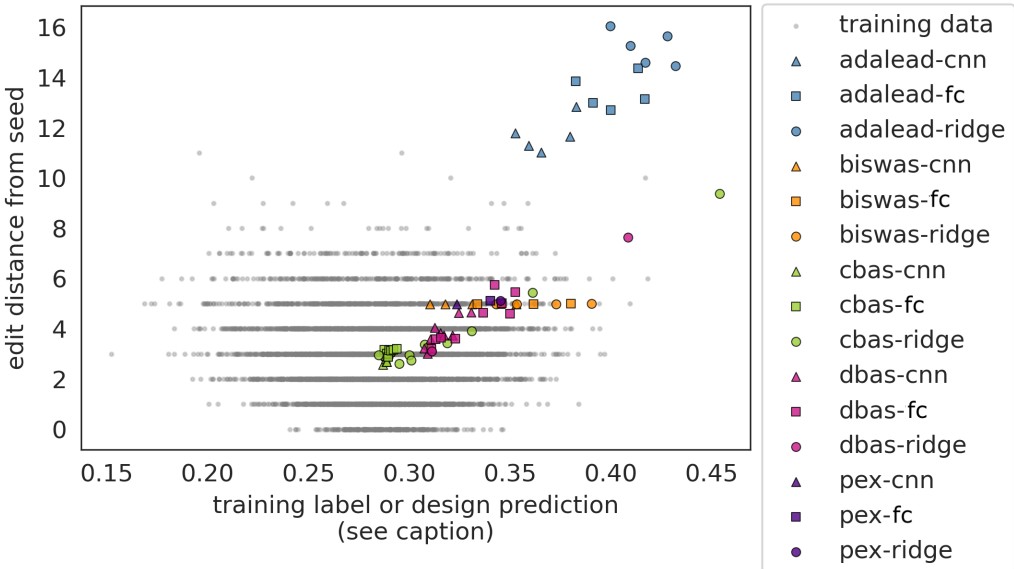

*Figure 9.* **Design algorithm configurations for designing RNA binders.** Gray dots give the label ($x$-axis) and edit distance from a seed sequence ($y$-axis) for the 5k training sequences used by the predictive models. Colored markers give the average design prediction ($x$-axis) and average edit distance from the seed ($y$-axis) for each configuration on the menu. Multiple markers of the same type correspond to configurations with the same design algorithm and predictive model, but different hyperparameter values. For example, the five blue triangles correspond to `AdaLead` using the convolutional ensemble and threshold hyperparameter values $\{0.2, 0.15, 0.1, 0.05, 0.01\}$.

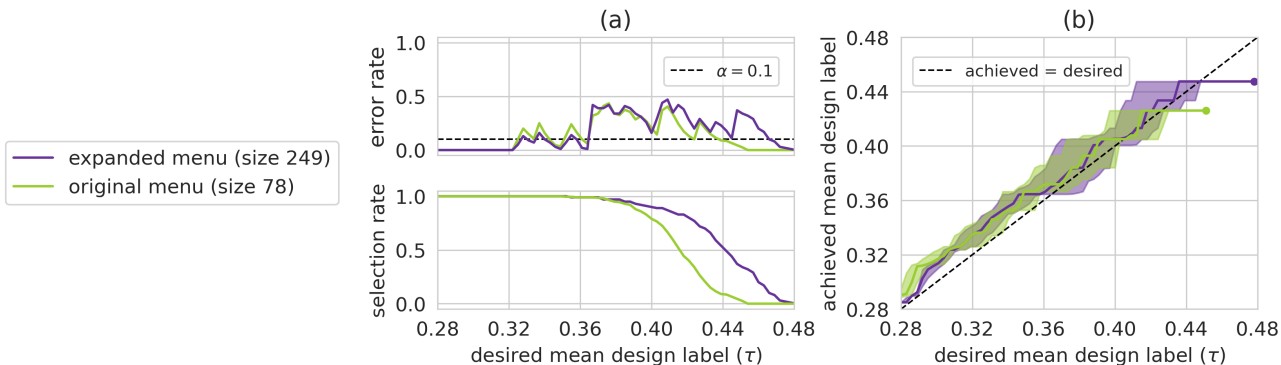

*Figure 10.* **Design algorithm selection for designing RNA binders (expanded menu).** (a) Error rate (top; lower is better) and selection rate (bottom; higher is better) for our original results (with menu of size 78; green) and our method with the expanded menu of size 249 diagrammed in Figure 8 (purple). (b) For both menus, the median (solid line) and 20th to 80th percentiles (shaded regions) of the lowest mean design label achieved by selected configurations, over trials for which the method did not return the empty set. Dots mark where each median trajectory ends (i.e., the value of $\tau$ beyond which the method ceased to select any configuration, and the lowest mean design label of configurations selected for that $\tau$). Results on or above the dashed diagonal line indicate that selected configurations are successful.

$Q \in \{0.1, 0.2\}$. Both `CbAS` and `DbAS` involve a generative model, which was a variational autoencoder (VAE) with 10 latent dimensions, and fully connected models with 20-unit hidden layers for both the encoder and decoder. Both algorithms were run for twenty iterations; each iteration retrained the VAE for five epochs using Adam with learning rate $10^{-3}$. `PEX` (Ren et al., 2022) was run with default values of all hyperparameters. The resulting menu of design algorithm configurations varied greatly in their mean design prediction, as well as their distance from the training sequences (Fig. 9).

To assess how the multiplicity correction might impact selection rates for larger menus, we also ran our method with an expanded menu of 279 configurations. In addition to the hyperparameter settings listed above, this menu included different values of the `AdaLead` recombination rate hyperparameter, $r \in \{0.1, 0.2, 0.5\}$, and numbers of hidden units ($\{5, 10\}$) and latent dimensions ($\{20, 100\}$) in the VAE used by both `CbAS` and `DbAS` (Fig. 8). Interestingly, our method exhibited similar error rates but higher selection rates with this larger menu, including non-zero selection rates for a broader range of $\tau$ than with the original menu of size 79 (Fig. 10a), due to the expanded menu containing more configurations that produce higher mean design labels.

### E.2   Density Ratio Estimation

To estimate the density ratio function, $p_{X;\lambda}(\cdot)/p_{\text{lab}}(\cdot)$, for every configuration on the menu, $\lambda \in \Lambda$, we used multinomial logistic regression-based density ratio estimation (MDRE) (Srivastava et al., 2023). MDRE builds upon a formal connection between density ratio estimation (DRE) and classification (Bickel et al., 2009; Gutmann & Hyvärinen, 2012): for the (correctly specified) binary classifier that minimizes the population cross-entropy risk in distinguishing between samples from two distributions, its logit for any input $x$ is equivalent to the density ratio for $x$, $p_{X;\lambda}(x)/p_{\text{lab}}(x)$ (Gutmann & Hyvärinen, 2012). In practice, however, we can only hope to find a classifier that minimizes the empirical risk between finite samples from the two distributions. This distinction between the population and empirical risks hinders DRE far more than it does classification for classification's sake: obtaining the density ratio requires getting the exact value of the population-optimal classifier's logit, whereas optimal classification performance can be achieved by any classifier that learns the same decision boundary as the optimal classifier, even if its logits differ from the optimal classifier. Accordingly, we found that for many configurations on the menu, classifiers with very low training and validation losses often yielded poor approximations of the density ratio. This was particularly true when the design and labeled distributions were far apart, because the classification problem is too easy given finite samples: many different classifiers can minimize the empirical risk, none of which may happen to coincide with the population-optimal one whose logits are equivalent to density ratios. Telescoping density ratio estimation (Rhodes et al., 2020) tackles this problem by constructing intermediate distributions that interpolate between the numerator and denominator distributions, creating a sequence of "harder" binary classification

problems for which there are fewer empirically optimal classifiers, and whose resulting estimated density ratios can be combined through a telescoping sum to approximate the original density ratio of interest. MDRE is similarly motivated but constructs a single multi-class classification problem between the intermediate distributions, justified by theoretical connections between the population-optimal classifier and the density ratio analogous to those described above (Srivastava et al., 2023). Concretely, let $h^c(x)$ denote the unnormalized log-probability according to a trained classifier that $x$ belongs to distribution $c \in \{1, \ldots, C\}$, where $C$ denotes the total number of distributions. MDRE uses $\exp(h^i(x) - h^j(x))$ to approximate the density ratio between distributions $i$ (numerator) and $j$ (denominator).

Note that the design algorithm selection problem naturally lends itself to the construction of the intermediate distributions used by MDRE, because many design algorithms have hyperparameters that navigate how far the design distribution strays from the labeled distribution. It is often of interest to include different settings of such hyperparameters on the menu, in which case all of these configurations' density ratios with the labeled distribution can be approximated using one MDRE classifier. For the RNA binder experiments, denote configurations by `[design algorithm]-[predictive model]-[hyperparameter value]`—for example, `Biswas-CNN-0.02` refers to running `Biswas` with the convolutional ensemble predictive model and temperature hyperparameter $T = 0.02$. We fit a separate MDRE model for each combination of a design algorithm and a predictive model—that is, a separate multi-class classifier for `AdaLead-ridge-*`, for `AdaLead-FC-*`, for `AdaLead-CNN-*`, for `Biswas-ridge-*`, for `Biswas-FC-*`, for `Biswas-CNN-*`, for `CbAS-ridge-*`, for `CbAS-FC-*`, for `CbAS-CNN-*`, for `DbAS-ridge-*`, for `DbAS-FC-*`, for `DbAS-CNN-*`, for `PEX-ridge`, for `PEX-FC`, and for `PEX-CNN`, where the last three reduced to binary classification problems. Each of these classifiers was fit on the 5k training sequences and $N = 50$k designed sequences from each included configuration. For example, the 10-category classifier for estimating density ratios for `CbAS-ridge-*` was fit on the 5k training sequences and 50k sequences each from `CbAS-ridge-0.1`, `CbAS-ridge-0.2`, ..., `CbAS-ridge-0.8`, and `CbAS-ridge-0.9`. Each classifier was a model with one 256-unit hidden layer and a quadratic final layer, trained for 100 epochs using Adam with a learning rate of $10^{-3}$.

