# OpenReview forum: "Reliable Algorithm Selection for Machine Learning-Guided Design"
_ICML.cc/2025/Conference — ICML 2025 poster_

### Official Review · Reviewer_pgJa · 2025-02-26

**Overall Recommendation:** 3

**Summary:**

This paper proposed a method for design algorithm selection which combines designs' predicted property values with held-out labeled data to reliably assess whether a candidate design algorithm configuration produces successful designs. Specifically, the method selects configurations for design algorithms such that the resulting outcome (label) satisfies a pre-defined success criterion (e.g., having a sufficient fraction of designs exceeding a certain threshold) with error rate at most $\alpha$.

To tackle this problem, the paper first formalizes the problem as hypothesis testing over a menu of candidate design algorithms, and then using prediction-powered inference techniques adapted to covariate shift (from held-out labeled data) to obtain statistically valid p-values for the hypothesis tests.

Under the assumption that the density ratio between the design and labeled distributions are known, the paper gives theoretical guarantee that the selected configurations achieves the success criterion for any error rate. Without known density ratio, the paper provides empirical evidence of the procedure by simulation on two biologically motivated examples that their method successfully controls the error rate while identifying configurations that produce high-quality designs.

**Claims And Evidence:**

The central claim of the paper is that the proposed algorithm (hypothesis testing procedure combined with prediction powered inference for valid p-values) is able to select design algorithm configurations that achieves the success criterion with guarantees on any error rate (i.e. probability of selecting algorithms that don't match the success criterion).

Under the assumption that the density ratio between the design and labeled distributions are known, Theorem 3.1 and Theorem A.1 give theoretical guarantees that using finite-sample or asymptotically valid p-values, the proposed procedure controls the error rate at any level $\alpha$.

When this assumption doesn't hold, the paper combines the approach with a multinomial logistic regression-based method for estimating the density ratio. Empirically, the paper compares their approach against other baselines (prediction-only model, GMMForecasts) and
shows that on the protein and RNA design tasks, their method outperforms others in selecting successful configurations.

**Essential References Not Discussed:**

The paper discusses their method within the context of machine-guided design and prediction-powered inference.
The overall problem formulation might also be related to the area of Bayesian optimization and active learning.

**Experimental Designs Or Analyses:**

Without known density ratio, the paper provides empirical evidence of the procedure by simulation on two biologically motivated examples that their method successfully controls the error rate while identifying configurations that produce high-quality designs.
In the experiments, the paper measures both the error rate of the selected configurations (probability of selecting at least one unsuccessful configuration) as well as the selection rate to ensure the algorithm is not overly conservative that outputs no configurations.

**Methods And Evaluation Criteria:**

The paper proposes to use a multiple hypothesis testing approach to test if each configuration fits the user-defined success criterion. To obtain statistically valid p-values for the hypothesis test, the paper uses prediction-powered inference adapted for covariate shift to extract information from these predictions without being misled by prediction error. It then usees a Bonferroni correction to bound the overall error rate.
The overall algorithm makes sense and gives guarantees under the assumption of known density ratio between design and labeled distributions.

**Other Comments Or Suggestions:**

n.a.

**Other Strengths And Weaknesses:**

n.a.

**Questions For Authors:**

- Algorithm 1 (line 6) selects p-values by a Bonferroni-style threshold. Would this possibly lead to overly conservative selections?
- Related, can you get theoretical guarantees on the selection rate?
- Do you have any theoretical guarantees if the density ratio is not known? Or, if the support of the densities are different? Are there possible problems that might arise in your analysis?

**Relation To Broader Scientific Literature:**

This work proposes the use of multiple hypothesis testing and prediction-powered inference applied to covariate shift to address machine learning-guided design problems where out-of-distribution predictions are common. Their proposed method gives a systematic way to choose design algorithms that fits the success criterion with guaranteed error rates.

**Theoretical Claims:**

I checked the proofs for the theoretical claims (Theorem 3.1 and Theorem A.1) and think it's correct.

---

> ### Author Rebuttal · Authors · 2025-04-01
>
> We appreciate your feedback! We would like to refer the reviewer to the first 3 paragraphs of our response to reviewer EEWq, as we believe there has been a misconception regarding our primary contributions.
>
> In particular, to our knowledge, our work is the first to formalize and propose a rigorous solution to the problem of design algorithm selection (DAS): a principled method for making the decisions that every practitioner of ML-guided design must make, and which might otherwise be biased by personal preference or convenience. None of the reviewers have suggested any alternatives, which testifies to the novelty and difficulty of DAS and the creativity of our solution.
>
> Our responses to specific points follow.
>
> - "Would [the Bonferroni correction] lead to overly conservative selection?" Empirically, this was not a problem. Our method had low error rates while maintaining high selection rates, compared to baselines (Figs. 3a,4a). In new results with a 5X-larger menu for the protein GB1 experiments ([Fig. R1](https://shorturl.at/qE6TG), see first comment to reviewer Bfc7 for details), our method actually has higher selection rates than with the original menu, as the expanded menu contains more configurations with higher mean design labels. If one is concerned about conservatism, however, a conceptual strength of our formalization of DAS as a multiple testing (MT) problem is that *any* MT procedure that controls FWER can be used instead of Bonferroni. E.g., one can use procedures that respect the hierarchical [1] or correlation structure between configurations [2], which may yield less conservative multiplicity corrections. Since our goal was to establish a first, general solution to DAS, we instantiated Alg. 1 with Bonferroni, as it doesn't require assumptions about how configurations are related.
>
> - "Can you get theoretical guarantees on the selection rate?" No, because if the practitioner specifies an impossible success criterion, then the selection rate must be zero. That is, no non-zero lower bound exists in general.
>
> - Potential issues with estimated density ratios (DRs): While the guarantees in the paper do not hold with DRE, this does not nullify the value of having guarantees when DRs are known, in contrast to baselines w/o guarantees at all. We appreciate your concern, however, and present new results in [Fig. R1](https://shorturl.at/qE6TG) showing that our method performs similarly in the protein GB1 experiments using estimated vs. known DRs. Both there and in our RNA experiments in Fig. 4, our method with DRE outperforms the baselines. Also note that DRE yields consistent estimators of the true DRs if the model class is correctly specified (flexible enough) [3], and is routinely used in ML applications involving importance weighting [4].
>
> To further address the concerns regarding Bonferroni and DRE, we also present new RNA experiments in [Fig. R2](https://shorturl.at/mOSna), which use DRE with an expanded menu of size 249 containing the hierarchical configuration space shown in [Fig. R3](https://shorturl.at/X0EOu). Our method performs well in these experiments.
>
> - Thank you for suggesting the relevance to Bayesian optimization and active learning! We now mention them in "Related Work," condensed here:
> Bayesian optimization (BO) is a paradigm for iteratively selecting designs, acquiring their labels, and updating a predictive model in order to optimize a property of interest. In each round, BO chooses the design that globally maximizes some acquisition function quantifying desirability based on the model's predictions. A typical goal of BO is to converge to the global optimum as the rounds progress, under regularity conditions [5]. In contrast, the goal of DAS is to achieve criteria on the distribution of design labels to be imminently proposed. Such guarantees can help justify designs to stakeholders when acquiring labels for even one round is resource-intensive, and the priority is to achieve specific criteria within one or a few rounds. However, nothing precludes BO from being used as design algorithms within our framework: configurations of BO with, e.g., different HPs or acquisition functions can be on the menu.
>
> Finally, we emphasize that our method empirically outperforms the baselines. Despite the impact of the DAS problem, no reviewer has suggested any alternative approaches. If the reviewer would like to suggest one, we are happy to include it!
>
> [1] Bretz et al. A graphical approach to sequentially rejective multiple test procedures. Stat. Med. 2009.
>
> [2] Dudoit et al. Multiple hypothesis testing in microarray experiments. Stat. Sci. 2003.
>
> [3] Gutmann & Hyvarinen. Noise-contrastive estimation of unnormalized statistical models. JMLR 2012.
>
> [4] Sugiyama et al. Density ratio estimation in machine learning. 2012.
>
> [5] Srinivas et al. Gaussian process optimization in the bandit setting. ICML 2010.

---

### Official Review · Reviewer_EEWq · 2025-03-06

**Overall Recommendation:** 2

**Summary:**

Hyperparameter tuning and algorithm selection can be really tricky in real scenes. This paper proposes a method based on prediction-powered inference techniques for design algorithm selection, aiming to choose some settings (configurations) that satisfy users' demands. Two practical experiments demonstrate this method can effectively select ideal settings.

**Claims And Evidence:**

Yes. The authors use p-values to support their design further.

**Essential References Not Discussed:**

No.

**Experimental Designs Or Analyses:**

Yes, both experiments seem sound to me.

**Methods And Evaluation Criteria:**

The method makes sense.

**Other Comments Or Suggestions:**

The reference in line 275 cannot jump.

**Other Strengths And Weaknesses:**

Strengths:
1. The idea of selecting reliable algorithms has a strong potential to benefit all machine learning communities.
2. The paper is well-written and easy to comprehend.

Weaknesses:
1. The major concern lies in that the authors have claimed a big idea, but the fact is that the menus used in both experiments only contain roughly 100 choices, which is far less in practical scenarios. A single method can require multiple hyperparameters to tune when applied to a new task, let alone we are in an era where new methods for a single task are merging every day.
2. The method is guaranteed by the assumption that the density ratios between the design and labeled data distributions are known, which can vary in real experiments or even be impossible to estimate.
3. The method uses held-out labeled data to characterize how prediction error affects the evaluation, which acquires lots extra information.

**Questions For Authors:**

1. What does '=' stands for in Figure 3(b) and Figure 4(b)?
2. Can your method be extended to more complicated tasks? For now, each of the RNA binder design algorithms contains one or zero hyperparameters, and all parameters are continuous. Is this method still satisfying when facing categorical parameters and methods with multiple parameters? Proving this would greatly help this paper to be more convincing.

**Relation To Broader Scientific Literature:**

The method can speed up the process of finding optimal hyperparameter and algorithm combinations and significantly reduce the cost of wet lab experiments.

**Theoretical Claims:**

I check the theoretical claims in the main context, showing their framework can guarantee a high success rate for selecting desirable settings.

---

> ### Author Rebuttal · Authors · 2025-04-01
>
> Thank you for your feedback! We are glad the reviewer found the idea of selecting reliable algorithms to have strong potential to benefit the ML community. We believe there has been a misconception regarding our primary contributions.
>
> **[First to formalize + address DAS]** To our knowledge, although ML has revolutionized the design of proteins, small molecules, and other modalities, this work is the first to formalize the problem of design algorithm selection (DAS) and propose a solution with any guarantees. None of the reviewers have suggested any baselines or alternative approaches, which testifies to the novelty and difficulty of DAS and the creativity of our solution.
>
> **[Not just HP tuning]** This work is not simply a variant of hyperparameter tuning as it is performed in supervised learning. Tuning cannot be performed for design tasks the same way, because we never have the labels of designs needed to evaluate each design algorithm configuration. This is why our major technical innovation is to rigorously evaluate which configurations will be successful, in the complete absence of labels for their designs.
>
> **[Formalized population-level success criteria w/ high-probability guarantees]** Another innovation is formalizing + achieving a common goal in practice: to produce a *population* of designs that satisfies a desired criterion (e.g., that at least 10% of designs' labels surpass some threshold, or that the average design label does so). Our method provides high-probability guarantees for a broad class of such population-level success criteria. Prior work has focused instead on uncertainty over individual designs, rather than population-level criteria often sought in practice.
>
> We are glad the reviewer found the paper well-written, and appreciate the concerns regarding the method's practicality, addressed below.
>
> - "Menus ... only contain roughly 100 choices": See [Fig. R1](https://shorturl.at/qE6TG) and [Fig. R2](https://shorturl.at/mOSna) for new protein and RNA experiments with menu sizes 501 and 249, respectively, where our method still performs well. However, even for the original smaller menus, our method outperformed the baselines. Are there alternative approaches the reviewer might propose? We are happy to try them.
>
> - "[Density ratios can be] impossible to estimate": Fig. 4 showed empirically that our method is still effective with a relatively simple classifier-based density ratio estimation (DRE) technique. Also see [Fig. R1](https://shorturl.at/qE6TG) for new results showing our method performs well with DRE in the protein GB1 experiments. Note that DRE is routinely used in ML applications involving importance weighting [1,2] and yields consistent estimators of the DRs if the model class is correctly specified (flexible enough) [3]. We agree that the guarantees don't hold with DRE, but this doesn't nullify the value of having guarantees when DRs are known, in contrast to baselines without guarantees at all.
>
> - Requires "held-out labeled data (HLD)": Figs. 3&4 compared to baselines that do not require HLD (prediction-only and GMMForecasts), which instead used *all* the labeled data to train the predictive model. Our method outperformed these baselines. We also note that the entire literature on post-hoc calibration for uncertainty quantification (including e.g. conformal prediction) requires HLD. Indeed, there is a fundamental limit to how calibrated models can be w/o using HLD [4]. In practice, collecting some extra labeled data can be a reasonable investment when the method's guarantees can help justify costly resources for synthesizing designs to project stakeholders. Lastly, our method is amenable to cross-fitting variants that do not completely hold out data [5] (happy to elaborate).
>
> - Can the method handle "categorical parameters and ... multiple parameters?": See [Fig. R2](https://shorturl.at/mOSna) for new RNA experiments with an expanded menu of size 249, containing the more complex hierarchical configuration space illustrated in [Fig. R3](https://shorturl.at/X0EOu). Our method performs well here (similar error rates with higher selection rates), as the expanded menu contains configurations with higher mean design labels than the original menu.
>
> - Fig 3B,4B: The "=" is an equals sign, indicating that the diagonal line is the y = x line. Results on/above the diagonal mean that selected configurations are successful.
>
> - Line 275: Thank you for noting the broken link! Now fixed.
>
> [1] Sugiyama et al. Density ratio estimation in machine learning. 2012.
>
> [2] Grover et al. Bias correction of learned generative models using likelihood-free importance weighting. NeurIPS 2019.
>
> [3] Gutmann & Hyvarinen. Noise-contrastive estimation of unnormalized statistical models. JMLR 2012.
>
> [4] Bengs et al. Pitfalls of epistemic uncertainty quantification through loss minimisation. NeurIPS 2022.
>
> [5] Zrnic & Candes. Cross-prediction-powered inference. PNAS 2024.

---

### Official Review · Reviewer_Bfc7 · 2025-03-13

**Overall Recommendation:** 4

**Summary:**

When performing model-guided design, the goal is to propose new objects x that have some desired property, where the relationship between x and the property is approximated by a predictive model f(x). The problem is that f(x) may be unreliably when proposing x far from the training data. This makes it difficult to choose among design algorithms. Each defines a distribution over potential designs. How do we estimate the performance of a design algorithm without actually measuring the property for these designs (which could involve an expensive wet-lab experiment)?

The authors propose an appealing frequentist formalism for the problem and leverage the recent 'prediction powered inference' framework to propose an algorithm with desirable properties.

**Claims And Evidence:**

The evaluation is a bit confusing at first, as it requires really understanding the frequentist formulation of the design algorithm selection problem. However, given this problem formulation, the way that algorithms are evaluated makes a lot of sense.

**Essential References Not Discussed:**

None

**Experimental Designs Or Analyses:**

Yes, I think it was posed correctly.

**Methods And Evaluation Criteria:**

The proposed method is well motivated and appealing. I appreciate that subtle variations are also provided for regimes where certain required quantities (the density ratio term) are note available.

I found the set of baselines to be very well presented and well motivated.

**Other Comments Or Suggestions:**

When I got to the experiments section, I was surprised to find that the 'menu' corresponded to a range of values for a single real-valued hyper-parameter. There is a lot of structure across this menu that the method doesn't exploit. The hyper-parameter range is also discretized at an arbitrary resolution of 100 values, but the menu size drives the performance of the algorithm due to the multiple testing correction. I worry that this choice of 100 could have had a big qualitative impact on the results.

For problems where the density ratio was tractable, it would have been helpful to see what the performance difference is between the case when using the true density ratio vs. an approximated one.

**Other Strengths And Weaknesses:**

I found figures 3A and 4A very confusing. I feel like there should be a better way to present this data that is mostly saturated at y = 0 or y = 1.

**Questions For Authors:**

Can you please show that the experiments' outcomes are not sensitive to the discretization resolution discussed above?


## After Authors' response ##
Thanks for addressing my comments. I think the proposed changes will strengthen the manuscript.

**Relation To Broader Scientific Literature:**

It does a good job of framing things in terms of related work

**Theoretical Claims:**

I do not have the required technical background to assess the correctness of the proofs in the appendix.

---

> ### Author Rebuttal · Authors · 2025-04-01
>
> We are grateful for the positive evaluation of our work, and are glad the reviewer found our method and experiments well-motivated and appealing. Our responses to specific comments and questions follow.
>
> - "The hyper-parameter range is also discretized at an arbitrary resolution of 100 values, but the menu size drives the performance of the algorithm due to the multiple testing correction." We appreciate your concern regarding this point. See [Fig. R1](https://shorturl.at/qE6TG) for new results where the menu for the protein GB1 experiments is discretized to contain 501 values (np.arange(0.2, 0.7, 0.001) instead of the original np.arange(0.2, 0.7, 0.005)). Our method still keeps error rates below the user-specified level, consistent with the guarantees. Interestingly, it does so with *higher* selection rates for greater values of tau (x-axis) than with the original smaller menu. That is, the multiple testing correction did not make the method more conservative with a 5X-larger menu; in fact, the method actually selected successful configurations more frequently, because the expanded menu contains more configurations with greater mean design labels.
>
> - "I was surprised to find that the 'menu' corresponded to a range of values for a single real-valued hyper-parameter. There is a lot of structure across this menu that the method doesn't exploit." Thank you for this insightful observation. Please see [Fig. R2](https://shorturl.at/mOSna) for new RNA experiments with an expanded menu of size 249, containing the more complex hierarchical configuration space illustrated in [Fig. R3](https://shorturl.at/X0EOu). Our method empirically performs well in these experiments, but we agree that it would be possible to better exploit the structure in the menu induced by the relationships between different configurations. This is a conceptual strength of formalizing design algorithm selection as a multiple testing problem: any multiple testing procedure that controls family-wise error rate (FWER) can actually be used in place of the Bonferroni correction in Line 6 of Alg. 1, meaning one can use FWER-controlling procedures that respect, e.g., the hierarchical organization of the configuration space [1] or the correlations between configurations [2]. We instantiated Alg. 1 with the Bonferroni correction for full generality, as it doesn't require any assumptions about how configurations are related, but replacing this with procedures that exploit these relationships is a great direction for future work. Note that using such structure-respecting multiple testing procedures would not improve the existing guarantees on the error rate, but might improve (i.e., increase) the selection rate by yielding a less conservative multiplicity correction.
>
> - "For problems where the density ratio was tractable, it would have been helpful to see what the performance difference is between the case when using the true density ratio vs. an approximated one." Thank you for this suggestion. See [Fig. R1](https://shorturl.at/qE6TG) for new results where we estimated the density ratios (DRs) for the protein GB1 experiments. Specifically, we separately estimated the labeled distribution and the design distribution corresponding to each configuration. For the former, we performed maximum-likelihood estimation with Laplace smoothing (with pseudocounts of 1) to estimate the site-specific categorical distributions using the held-out labeled sequences; for the latter, we did the same using the design sequences. For a given sequence, we then took the ratio of its densities under these two estimated distributions as the estimated DR. The results from our method using these estimated DRs are very similar to the original results using the known DRs.
>
> - Figs. 3A, 4A: Thank you for noting that the presentation was confusing. To clarify, would the reviewer prefer that we zoom-in the y-axes of plots closer to 0 for error rate and 1 for selection rate? Note that the methods have error rates and selection rates that span the entire [0, 1] range, because we wanted to thoroughly compare their performance across a wide range of success criteria (x-axis values) that practitioners might be interested in. However, we have zoomed-in the error rate plot of [Fig. R1](https://shorturl.at/qE6TG) closer to 0 as suggested, since the plotted rates are all less than 0.1.
>
> [1] Bretz et al. A graphical approach to sequentially rejective multiple test procedures. Stat. Med. 2009.
>
> [2] Dudoit et al. Multiple Hypothesis Testing in Microarray Experiments. Stat. Sci. 2003.

---

### Decision · Program_Chairs · 2025-05-01

**Decision:**

Accept (poster)

**Comment:**

This paper studies a pretty neat problem: how do we actually select optimization/design algorithms without actually making many more measurements of the objective function? The authors present a frequentist approach to this problem with guarantees under the assumption that the density ratio between the design and labeled distributions are known. The authors responded comprehensively to the questions raised by the reviewers during the author feedback period. I generally agree with the authors that Reviewer EEWq appears to have overlooked the main contributions of this work, and that the setting considered here is pretty substantively different from hyperparameter tuning (although I suppose could reduce to hyperparameter tuning in the setting where the set of design algorithms differed only by hyperparameter values?)